

# Effects of radial radio-frequency field inhomogeneity on MAS solid-state NMR experiments

Kathrin Aebischer[1], Zdeněk Tošner[2], and Matthias Ernst[1]

[1]Physical Chemistry, ETH Zürich, Vladimir-Prelog-Weg 2, 8093 Zürich, Switzerland
[2]Department of Chemistry, Faculty of Science, Charles University, Hlavova 8, 12842 Prague 2, Czech Republic

**Correspondence:** Matthias Ernst (maer@ethz.ch)

**Abstract.** Radio-frequency field inhomogeneity is one of the most common imperfections in NMR experiments. They can lead to imperfect flip angles of applied radio-frequency (rf) pulses or to a mismatch of resonance conditions resulting in artifacts or degraded performance of experiments. In solid-state NMR under magic-angle spinning, the radial component becomes time-dependent because the rf-irradiation amplitude and phase is modulated with integer multiples of the spinning frequency.

5   We analyze the influence of such time-dependent MAS-modulated rf fields on the performance of some commonly used building blocks of solid-state NMR experiments. This analysis is based on analytical Floquet calculations as well as numerical simulations taking into account the time dependence of the rf field. We find that compared to the static part of the rf-field inhomogeneity, such time-dependent modulations play a very minor role in the performance degradation of the investigated typical solid-state NMR experiments.

## 1   Introduction

Radio-frequency (rf) field inhomogeneity describes the spatial inhomogeneity of the rf-field inside the coil or sample volume and is one of the major experimental imperfections that lead to artifacts or reduced efficiency in NMR experiments. The magnitude of the rf-field amplitude distribution over the sample space can be estimated with a nutation experiment (Torrey,

15   1949; Barnaal and Lowe, 1963). Measuring such nutation spectra of thin sample slices placed along the rotor axis allows the characterization of the spatial rf-field distribution along the coil axis (Nishimura et al., 2001; Paulson et al., 2004). The full spatial distribution, however, is only accessible using gradient methods (Guenneugues et al., 1999; Odedra and Wimperis, 2013) that are typically not available in solid-state NMR probes. Alternative approaches include the measurement of the rf-field amplitude using the ball-shift experiment (Maier and Slater, 1952), numerical simulations based on finite elements or

20   approximative analytical solutions of the Maxwell equations (Engelke, 2002; Tošner et al., 2017, 2018). The design of the coil geometry has a major influence on the magnitude and distribution of the rf-field amplitude over the active sample volume and different geometries have been proposed to improve the rf homogeneity (Idziak and Haeberlen, 1982; Privalov et al., 1996;





Li et al., 2006). However, in solid-state NMR probes, solenoid coils along the sample spinning axis are most commonly used due to the high achievable rf-field amplitudes. The gap between the rotor and the coil is minimized in order to optimize the filling factor. This design choice typically leads to large rf-field inhomogeneity that can manifest itself in reduced efficiency in experiments such as cross polarization (Hartmann and Hahn, 1962; Stejskal et al., 1977), homonuclear decoupling (Bielecki et al., 1989, 1990; Mote et al., 2016), heteronuclear decoupling (Purusottam et al., 2015; Frantsuzov et al., 2017), symmetry-based recoupling sequences (Levitt, 2007), or even pulsed recoupling experiments like REDOR (Nishimura et al., 2001).

Reducing the magnitude of the rf-field inhomogeneity can be achieved experimentally by physically restricting the sample along the rotor axis or even to a sphere in the center of the rotor (Lindon et al., 2009). Alternatively, gradients can be used for a virtual sample restriction (Charmont et al., 2000) but since gradients are not very common in solid-state NMR probes, this approach is rarely used. Another possibility are radio- or nutation-frequency selective pulses that can be used for the same purpose (Charmont et al., 2002; Aebischer et al., 2020). All these methods, however, are accompanied by a reduction in signal due to the restriction of the measured sample volume to a smaller part of the coil volume.

In solid-state NMR under magic-angle spinning (MAS) conditions, the radial component of the rf field gets modulated by time (Tekely and Goldman, 2001; Goldman and Tekely, 2001; Tošner et al., 2017) leading to further potential complications in the experiments. Such MAS-induced time-dependent radio-frequency fields could give rise to additional or modified resonance conditions or to other changes in the effective Hamiltonians generated by the pulse sequence. Besides the appearance of additional sidebands in cross-polarization experiments (Tekely and Goldman, 2001; Goldman and Tekely, 2001), nutation spectra (Elbayed et al., 2005) and reported phase distortions and loss of magnetization in MLEV16 sequences under MAS (Piotto et al., 2001), there have been very few studies of the effects of such modulations of the amplitude and phase of the rf field caused by MAS rotation. These modulations have been included in the design of a heteronuclear polarization transfer scheme based on optimal-control strategies (Tošner et al., 2017, 2018), where impressive gains have been shown. These improvements prompted us to investigate potential effects of such MAS modulated radio-frequency field amplitudes and phases on basic building blocks in common solid-state NMR pulse sequences in more detail. The approach we have chosen is rather simple. We use analytical approaches based on Floquet theory (Leskes et al., 2010; Scholz et al., 2010) and numerical simulations based on computed rf-field distributions in a typical MAS rotor to characterize the time evolution of the density operator under MAS rotation with and without time-dependent rf-field amplitudes and phases. The computational approach allows us to investigate the time evolution of the density operator in different spatial parts of the rotor and to magnify the amplitude or phase modulations of the rf field to obtain a better picture of their importance.

## 2 Radio-Frequency Fields in Solenoid Coils

To illustrate the magnitude and distribution of the rf-field amplitude and phase over the active sample volume in some typical MAS NMR probes, rf-field distributions were calculated based on Engelke (2002); Tošner et al. (2017). Figure 1 shows the relative amplitude and phase of the rf field in a cylindrical coordinate system as a function of $z$ (the axis along the rotor axis)





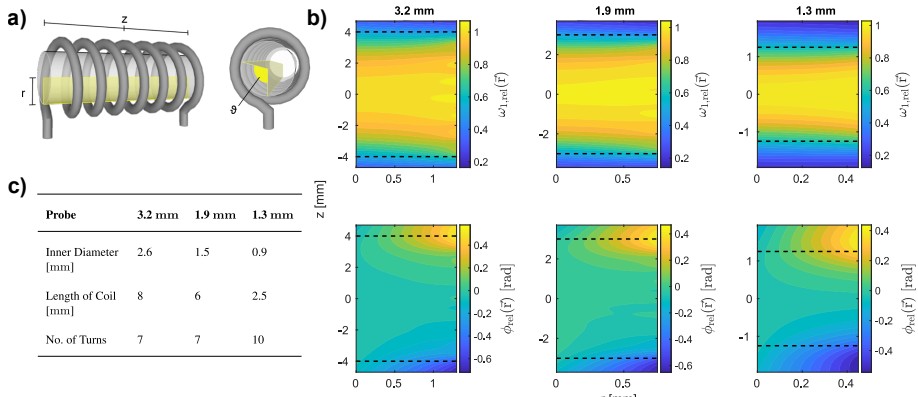

**Figure 1.** a) Coil geometry of a typical 3.2 mm Bruker MAS probe. The position within the sample space is indicated by the cylindrical coordinates $r$, $z$ and $\vartheta$. b) Spatial rf-field distributions for typical probe designs with MAS rotors of 3.2, 1.9 and 1.3 mm outer diameter at a frequency of 600 MHz. Relative rf amplitudes $\omega_{rel}$ and phases $\phi_{rel}$ are shown as a function of the position within the active sample volume for $\vartheta = 90°$. The length and position of the solenoid coil is indicated by dashed lines. c) Parameters of coil geometries of the 3.2, 1.9 and 1.3 mm MAS probes considered in this work. The $z$ values were sampled in steps of 0.05 mm for all three probes. The $r$ values were sampled in steps of 0.05 mm for the 3.2 and the 1.9 mm probes and in steps of 0.025 mm for the 1.3 mm probe. The sample volume considered in the numerical simulations and Floquet analyses presented in Section 5 was restricted to the length of the coil.

and $r$ (the radial direction) for three common probe designs with MAS rotors of 3.2, 1.9 and 1.3 mm outer diameter. In these plots, the angle $\vartheta = 90°$ was chosen. The maximum intensity of the rf amplitude distribution was used as the reference point for the relative amplitude $\omega_{rel}(\mathbf{r})$, a value of 1 hence means that the amplitude experienced at this position corresponds to the nominal rf amplitude. The rf phase was computed relative to the center point of the rotor with $(r, z) = (0, 0)$. One can clearly

see the decay of the rf amplitude towards the edges of the rotor along the rotor axis (large $z$ values) while phase errors mostly occur for large $r$ and $z$ values.

The radial dependence of the relative rf-field amplitude and phase as a function of the angle $\vartheta$ is shown in Fig. 2 for different values of $z$ and $r$ for the 3.2 mm MAS probe. Under sample rotation, the angle $\vartheta$ varies as a function of time and the rf-field

amplitude and phase are periodically modulated with the rotor frequency. The trajectories in Fig. 2 clearly show that the magnitude of these amplitude and phase modulations increases towards the edges of the rotor. We, therefore, expect crystallites located at large $r$ and $z$ values to experience the strongest modulations of the rf-field amplitude and phase.

For numerical simulations of spin dynamics, the numerical values for $\omega_{1,rel}(\mathbf{r})$ and $\phi_{rel}(\mathbf{r})$ obtained from simulations of

70 the rf-field distribution shown in Figs. 1 and 2 were used directly as input. For analytical calculations based on Floquet theory, a parametrisation of the values using a Fourier series with the MAS frequency, $\omega_r$, is more convenient and was obtained by





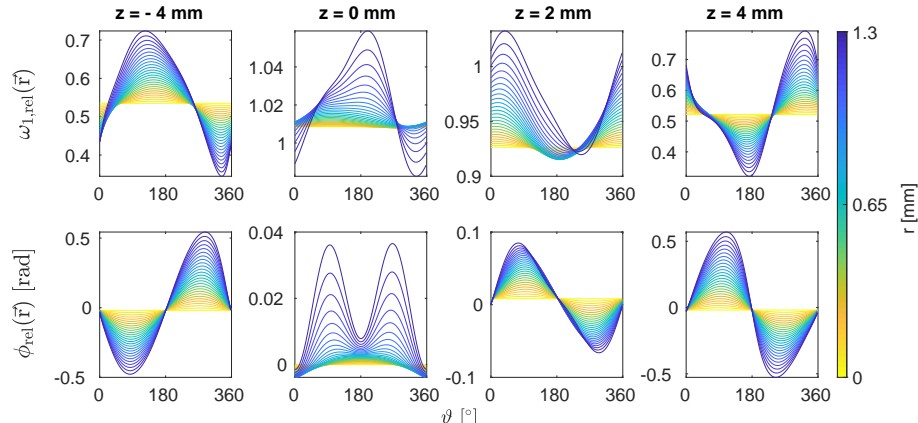

**Figure 2.** Relative rf amplitude $\omega_{1,\mathrm{rel}}$ and phase $\phi_{\mathrm{rel}}$ as a function of $\vartheta$ for $z = $ -4, 0, 2 and 4 mm and all $r$ values in the 3.2 mm MAS probe at a resonance frequency of 600 MHz. Under MAS, the rf amplitude and phase experienced by a crystallite will vary periodically with the rotor frequency. These time-dependent modulations of the rf-field are strongest at the edges of the rotor for large $r$ and $z$ values.

fitting the expressions

$$\omega_{\mathrm{rel}}(t) = A_0^{(\mathrm{A})} + \sum_{n=1}^{\infty} A_n^{(\mathrm{A})} \cdot \cos(n\omega_{\mathrm{r}}t + \phi_n^{(\mathrm{A})}),\tag{1}$$

$$\phi_{\mathrm{rel}}(t) = A_0^{(\mathrm{P})} + \sum_{n=1}^{\infty} A_n^{(\mathrm{P})} \cdot \cos(n\omega_{\mathrm{r}}t + \phi_n^{(\mathrm{P})})\tag{2}$$

to the amplitude and phase changes of the rf field (see Fig. 2). Typically, terms of the Fourier series up to $n = 4$ were used in the fits to characterize the time-dependent amplitude $\omega_{\mathrm{rel}}(t)$ and phase $\phi_{\mathrm{rel}}(t)$.

## 3   Theory

### 3.1   Floquet Description

In the high-field approximation the total Hamiltonian in the rotating frame under MAS for a homonuclear spin system com-
prised of $N$ $I$ spins is given by

$$\hat{\mathcal{H}}(t) = \sum_{n=-2}^{2} \sum_{p=1}^{N} \omega_p^{(n)} e^{in\omega_{\mathrm{r}}t} \hat{I}_z + \sum_{p<q} \sum_{\substack{n=-2 \\ n\neq 0}}^{2} \omega_{pq}^{(n)} e^{in\omega_{\mathrm{r}}t} \left[ 3\hat{I}_{pz}\hat{I}_{qz} - \hat{\boldsymbol{I}}_p \cdot \hat{\boldsymbol{I}}_q \right] + \sum_{p<q} \omega_{pq}^{(0)} \hat{\boldsymbol{I}}_p \cdot \hat{\boldsymbol{I}}_q + \hat{\mathcal{H}}_{\mathrm{rf}}(t).\tag{3}$$



Fourier components of the spatial tensors $\omega^{(n)}$ of the chemical shift of a spin $I_p$ and couplings between two spins $I_p$ and $I_q$ are given by

$$\omega_p^{(0)} = \Omega_p \tag{4}$$

$$\omega_p^{(n)} = \frac{2}{\sqrt{6}} d_{n,0}^2(\theta_{\mathrm{m}}) e^{-in\gamma} \sum_{m=-2}^{2} d_{m,n}^2(\beta) e^{-im\alpha} \rho_{2,m}^{(p)} \tag{5}$$

$$\omega_{pq}^{(0)} = 2\pi J_{pq} \tag{6}$$

$$\omega_{pq}^{(n)} = \frac{1}{\sqrt{6}} d_{n,0}^2(\theta_{\mathrm{m}}) e^{-in\gamma} d_{0,n}^2(\beta) \rho_{2,0}^{(pq)} \tag{7}$$

for the isotropic and anisotropic chemical shifts, the scalar $J$ and the anisotropic dipolar coupling. The sets of Euler angles $(\alpha, \beta, \gamma)$ describe the orientation of the tensors in the rotor-fixed frame and $d_{mm'}^\ell(\beta)$ denote the reduced Wigner matrix elements. The elements of the tensor in its principal axis system are denoted by $\rho_{lm}$. The most general rf Hamiltonian for Eq. (3) is given by

$$\hat{\mathcal{H}}_{\mathrm{rf}}(t) = \omega_1(t) \sum_{p=1}^{N} \left( \cos\phi(t) \hat{I}_{px} + \sin\phi(t) \hat{I}_{py} \right), \tag{8}$$

where both the rf amplitude $\omega_1(t)$ and the rf phase $\phi(t)$ can be time-dependent due to the irradiation scheme. Under MAS, the radial part of the rf inhomogeneity will lead to additional modulations of $\omega_1(t)$ and $\phi(t)$ that are periodic with the rotor frequency.

The system Hamiltonian can be transformed into an interaction frame with respect to $\hat{\mathcal{H}}_{\mathrm{rf}}$ by

$$\hat{\tilde{\mathcal{H}}}(t) = \hat{U}_{\mathrm{rf}}^{-1}(t) \hat{\mathcal{H}}(t) \hat{U}_{\mathrm{rf}}(t). \tag{9}$$

The propagator characterizing the interaction-frame transformation is given by

$$\hat{U}_{\mathrm{rf}}(t) = \hat{T} \exp\left( -i \int_0^t \hat{\mathcal{H}}_{\mathrm{rf}}(t') dt' \right), \tag{10}$$

where $\hat{T}$ is the Dyson time-ordering operator (Dyson, 1949) that ensures proper time-ordering of non-commuting operators in products. If the rf irradiation is periodic with $\tau_{\mathrm{m}}$, the $\hat{I}_z$ spin operators in Eq. (3) will transform according to

$$\hat{\tilde{I}}_z(t) = \sum_{\chi=x,y,z} a_\chi(t) \hat{I}_\chi = \sum_\chi \sum_k \sum_\ell a_\chi^{(k,\ell)} e^{ik\omega_{\mathrm{m}} t} e^{i\ell\omega_{\mathrm{eff}} t} \hat{I}_\chi, \tag{11}$$

where the transformation behaviour of the Cartesian spin operator $\hat{I}_z$ is characterized by the Fourier coefficients $a_\chi^{(k,\ell)}$ (Scholz et al., 2009, 2010). These coefficients only depend on the rf irradiation scheme and are independent of the spin system details. Generally, the interaction-frame trajectories of the spin operators are characterized by two basic frequencies, the modulation frequency of the pulse sequence $\omega_{\mathrm{m}} = \frac{2\pi}{\tau_{\mathrm{m}}}$ and an additional effective nutation frequency that can be determined from the overall flip angle over one period of the pulse scheme $\omega_{\mathrm{eff}} = \frac{\beta_{\mathrm{eff}}}{\tau_{\mathrm{m}}}$. This additional effective field is zero if the propagator over a full cycle of the pulse sequence is unity (Tan et al., 2016).



The interaction-frame Hamiltonian can thus be expanded as a Fourier series with three basic frequencies

$$\hat{\bar{\mathcal{H}}}(t) = \sum_n \sum_k \sum_\ell \hat{\mathcal{H}}^{(n,k,\ell)} e^{in\omega_{\mathrm{r}} t} e^{ik\omega_{\mathrm{m}} t} e^{i\ell\omega_{\mathrm{eff}} t} , \tag{12}$$

with Fourier components $\hat{\mathcal{H}}^{(n,k,\ell)}$

$$\hat{\bar{\mathcal{H}}}^{(n,k,\ell)} = \sum_{p=1}^N \omega_p^{(n)} \sum_\chi a_\chi^{(k,\ell)} \hat{I}_{p\chi} + \left[ \sum_{p<q} \omega_{pq}^{(0)} \hat{\boldsymbol{I}}_p \cdot \hat{\boldsymbol{I}}_q \right] \cdot \delta_{n,0} \cdot \delta_{k,0} \cdot \delta_{\ell,0}$$
$$+ \left[ \sum_{p<q} 3 \cdot \omega_{pq}^{(n)} \sum_\mu \sum_\chi a_{\mu\chi}^{(k,\ell)} \hat{I}_{p\mu} \hat{I}_{q\chi} - \left( \hat{\boldsymbol{I}}_p \cdot \hat{\boldsymbol{I}}_q \right) \cdot \delta_{k,0} \cdot \delta_{\ell,0} \right] \cdot (1 - \delta_{n,0}) , \tag{13}$$

where $\delta_{m,m'}$ denotes the Kronecker delta. For convenience, the general two-spin Fourier coefficients $a_{\chi\mu}^{(k,\ell)}$ were defined. They can be computed as the convolution of single-spin coefficients

$$a_{\mu\chi}^{(k,\ell)} = \sum_{k_1} \sum_{\ell_1} a_\mu^{(k_1,\ell_1)} a_\chi^{(k-k_1,\ell-\ell_1)} . \tag{14}$$

The scalar product of the $I$ spin vector operators remains time invariant under rf irradiation and can be incorporated into the $a_{\mu\mu}^{(0,0)}$ Fourier coefficients.

In triple-mode Floquet theory, the first-order effective Hamiltonian is given by all contributions that satisfy the resonance condition

$$n_0 \omega_{\mathrm{r}} + k_0 \omega_{\mathrm{m}} + \ell_0 \omega_{\mathrm{eff}} = 0 , \tag{15}$$

and thus the sum of non-resonant ($n_0 = k_0 = \ell_0 = 0$) and resonant terms

$$\hat{\bar{\mathcal{H}}}_{\mathrm{eff}}^{(1)} = \hat{\bar{\mathcal{H}}}^{(0,0,0)} + \sum_{n_0,k_0,\ell_0} \hat{\bar{\mathcal{H}}}^{(n_0,k_0,\ell_0)} . \tag{16}$$

Analogously, the second-order effective Hamiltonian is given by

$$\hat{\bar{\mathcal{H}}}_{\mathrm{eff}}^{(2)} = \hat{\bar{\mathcal{H}}}_{(2)}^{(0,0,0)} + \sum_{n_0,k_0,\ell_0} \hat{\bar{\mathcal{H}}}_{(2)}^{(n_0,k_0,\ell_0)} , \tag{17}$$

where

$$\hat{\bar{\mathcal{H}}}_{(2)}^{(n_0,k_0,\ell_0)} = -\frac{1}{2} \sum_{\nu,\kappa,\lambda} \frac{\left[ \hat{\bar{\mathcal{H}}}^{(n_0-\nu,k_0-\kappa,\ell_0-\lambda)}, \hat{\bar{\mathcal{H}}}^{(\nu,\kappa,\lambda)} \right]}{\nu\omega_{\mathrm{r}} + \kappa\omega_{\mathrm{m}} + \lambda\omega_{\mathrm{eff}}} . \tag{18}$$

The summation is restricted to values of $\nu$, $\kappa$ and $\lambda$ for which $\nu\omega_{\mathrm{r}} + \kappa\omega_{\mathrm{m}} + \lambda\omega_{\mathrm{eff}} \neq 0$ is satisfied in order to avoid singularities.



### 3.1.1 Theoretical Description Including RF Inhomogeneity

For spatial rf-field distributions that do not have cylindrical rotation symmetry, MAS will lead to a periodic modulation of the rf-field amplitude and phase experienced by a spin packet. At a given position, the general rf Hamiltonian including these additional modulations can be expressed as

$$\hat{\mathcal{H}}_{\mathrm{rf}}(t) = \omega_{1,\mathrm{rel}}(t) \cdot \omega_{1,\mathrm{nom}}(t) \left( \cos(\phi_{\mathrm{nom}}(t) + \phi_{\mathrm{rel}}(t)) \, \hat{I}_x + \sin(\phi_{\mathrm{nom}}(t) + \phi_{\mathrm{rel}}(t)) \, \hat{I}_y \right), \tag{19}$$

where $\omega_{1,\mathrm{nom}}(t)$ and $\phi_{\mathrm{nom}}(t)$ correspond to the nominal rf amplitude and phase, i.e., to the values corresponding to a perfectly homogeneous rf field. They are determined by the pulse scheme under investigation and are periodic with $\omega_{\mathrm{m}} = \frac{2\pi}{\tau_{\mathrm{m}}}$. Deviations from these nominal values are introduced by the relative rf amplitude and phase $\omega_{1,\mathrm{rel}}(t)$ and $\phi_{\mathrm{rel}}(t)$ that are periodic with $\omega_{\mathrm{r}} = \frac{2\pi}{\tau_{\mathrm{r}}}$. The overall period of the rf Hamiltonian will only be of finite length if the modulation frequency of the pulse scheme $\omega_{\mathrm{m}}$ and the rotor frequency $\omega_{\mathrm{r}}$ are commensurate

$$\omega_{\mathrm{m}} = \frac{\omega_{\mathrm{r}}}{c}, \tag{20}$$

where $c$ corresponds to the number of rotor cycles required for the synchronization of the rf irradiation and the MAS rotation. Such a synchronization condition is generally fulfilled for rotor-synchronized pulse schemes such as most recoupling sequences (Nielsen et al., 2012). For irradiation schemes that are typically applied asynchronously to avoid resonance conditions, a careful selection of the synchronization condition is required. In principle, the treatment can be generalized to all cases where the largest common divisor of $\omega_{\mathrm{r}}$ and $\omega_{\mathrm{m}}$ is not too small. Such a synchronization is required to make the modulations of the rf-field Hamiltonian by MAS cyclic over the basic repetition time of the sequence. The interaction-frame Hamiltonian can then be written as

$$\hat{\hat{\mathcal{H}}}(t) = \sum_{n=-2}^{2} \sum_{k} \sum_{\ell} \hat{\hat{\mathcal{H}}}^{(n,k,\ell)} \underbrace{e^{in\omega_{\mathrm{r}}t}}_{\text{MAS}} \underbrace{e^{ik\omega_{\mathrm{m}}t} e^{i\ell\omega_{\mathrm{eff}}t}}_{\text{rf irradiation}} \tag{21}$$

$$= \sum_{n=-2}^{2} \sum_{k} \sum_{\ell} \hat{\hat{\mathcal{H}}}^{(n,k,\ell)} e^{i(cn+k)\omega_{\mathrm{m}}t} e^{i\ell\omega_{\mathrm{eff}}t} \tag{22}$$

$$= \sum_{n'} \sum_{\ell} \hat{\hat{\mathcal{H}}}^{(n',\ell)} e^{in'\omega_{\mathrm{m}}t} e^{i\ell\omega_{\mathrm{eff}}t}, \tag{23}$$

where the substitution $n' = c \cdot n + k$ was used. The summation over the index $k$ runs from $-\infty$ to $+\infty$, the sum over $n'$ in Eq. (23) is thus also unrestricted. The resulting interaction-frame Hamiltonian is modulated with $\omega_{\mathrm{r}} = c \cdot \omega_{\mathrm{m}}$ due to the time dependence of the spatial part of the Hamiltonian during MAS (Fourier number $n$). The radial rf inhomogeneity leads to an additional modulation of the spin part with the rotor frequency. As the modulation frequency of the pulse sequence is commensurate with $\omega_{\mathrm{r}}$ these two modulations can be combined and a single Fourier number $n'$ can be used. Therefore, the Hamiltonian is given by a Fourier series with only two basic frequencies ($\omega_{\mathrm{m}}$ and $\omega_{\mathrm{eff}}$) and the triple-mode Floquet analysis is reduced to a bimodal treatment. Alternatively, one can continue with the triple-mode Floquet description and assume resonance conditions between $\omega_{\mathrm{m}}$ and $\omega_{\mathrm{r}}$. For resonant phenomena, the triple-mode Floquet approach is more suited since the additional





effective field can lead to changes in the resonance conditions leading to asynchronous sequences (Hellwagner et al., 2017; Tan
et al., 2015). For non-resonant phenomena, both descriptions will give equivalent results. The periodic rf amplitude and phase
modulations due to the radial inhomogeneity merely affect the interaction-frame transformation which is fully characterized
by $\omega_\mathrm{m}$ and $\omega_\mathrm{eff}$ as long as these rf-field modulations are periodic on the length of the interaction-frame trajectory.

Possible resonance conditions for the bimodal interaction-frame Hamiltonian of Eq. (23) are given by

$$n_0' \omega_\mathrm{m} + \ell_0 \omega_\mathrm{eff} = 0 \,. \tag{24}$$

As effective fields due to the rf inhomogeneity will typically be small compared to $\omega_\mathrm{r}$ and $\ell$ is limited to a maximum value of
two, it is reasonable to assume that only non-resonant terms will contribute to the effective Hamiltonian. To second order $\hat{\bar{\mathcal{H}}}_\mathrm{eff}$
can thus be written as

$$\hat{\bar{\mathcal{H}}}_\mathrm{eff} = \hat{\bar{\mathcal{H}}}^{(0,0)} - \frac{1}{2} \sum_{\nu,\lambda} \frac{\left[ \hat{\bar{\mathcal{H}}}^{(-\nu,-\lambda)}, \hat{\bar{\mathcal{H}}}^{(\nu,\lambda)} \right]}{\nu \omega_\mathrm{m} + \lambda \omega_\mathrm{eff}} \,, \tag{25}$$

where the summation over $\nu$ and $\lambda$ is restricted to values satisfying $\nu \omega_\mathrm{m} + \lambda \omega_\mathrm{eff} \neq 0$.

Each of the $\hat{\bar{\mathcal{H}}}^{(n',\ell)}$ Fourier components in Eq. (23) is composed of a sum of several $\hat{\bar{\mathcal{H}}}^{(n,k,\ell)}$ terms since there are multiple
combinations of $n$ and $k$ resulting in the same $n'$. As the index $n$ is limited to values between $\pm 2$ (limited by the rank of the
spatial tensor), the $\hat{\bar{\mathcal{H}}}^{(n',\ell)}$ are given by

$$\hat{\bar{\mathcal{H}}}^{(n',\ell)} = \sum_{n=-2}^{2} \hat{\bar{\mathcal{H}}}^{(n,n'-c\cdot n,\ell)} \,. \tag{26}$$

To first order, the non-resonant contribution to the effective Hamiltonian is simply given by the $\hat{\bar{\mathcal{H}}}^{(0,0)}$ Fourier component

$$\hat{\bar{\mathcal{H}}}^{(0,0)} = \hat{\bar{\mathcal{H}}}^{(0,0,0)} + \sum_{\substack{n=-2 \\ n\neq 0}}^{2} \hat{\bar{\mathcal{H}}}^{(n,-c\cdot n,0)}$$

$$= \sum_{p=1}^{N} \sum_{\chi} \sum_{n=-2}^{2} \omega_p^{(n)} a_\chi^{(-c\cdot n,0)} \hat{I}_{p\chi} + \sum_{p<q} \sum_{\chi,\mu} \sum_{\substack{n=-2 \\ n\neq 0}}^{2} 3 \cdot \omega_{pq}^{(n)} a_{\chi\mu}^{(-c\cdot n,0)} \hat{I}_{p\chi} \hat{I}_{q\mu} \,. \tag{27}$$

In full analogy to Tan et al. (2016), the second-order effective Hamiltonian can be decomposed into the three contributions
from commutator cross-terms between chemical-shift and dipolar-coupling terms

$$\hat{\bar{\mathcal{H}}}_{(2)}^{(0,0)} = \hat{\bar{\mathcal{H}}}_{\mathrm{I}\otimes\mathrm{I}} + \hat{\bar{\mathcal{H}}}_{\mathrm{I}\otimes\mathrm{II}} + \hat{\bar{\mathcal{H}}}_{\mathrm{II}\otimes\mathrm{II}} \,, \tag{28}$$



**MAGNETIC RESONANCE**
Open Access Discussions

where

$$\hat{\bar{\mathcal{H}}}_{\mathrm{I}\otimes\mathrm{I}} = \sum_{p=1}^{N}\sum_{n_1=-2}^{2}\sum_{n_2=-2}^{2}\sum_{\chi}\frac{-i}{2}\omega_p^{(n_1)}\omega_p^{(n_2)}q_\chi^{(n_1,n_2)}\hat{I}_{p\chi}, \tag{29}$$

$$\hat{\bar{\mathcal{H}}}_{\mathrm{I}\otimes\mathrm{II}} = \sum_{p\neq q}\sum_{\chi,\mu}\sum_{n_1=-2}^{2}\sum_{\substack{n_2=-2\\n_2\neq0}}^{2}\frac{-3i}{2}\omega_p^{(n_1)}\omega_{pq}^{(n_2)}q_{\chi,\mu}^{(n_1,n_2)}\hat{I}_{p\chi}\hat{I}_{q\mu}, \tag{30}$$

$$\hat{\bar{\mathcal{H}}}_{\mathrm{II}\otimes\mathrm{II}} = \sum_{p\neq q}\sum_{\chi}\sum_{n_1,n_2}-\frac{9i}{8}\omega_{pq}^{(n_1)}\omega_{pq}^{(n_2)}p_\chi^{(n_1,n_2)}\hat{I}_{p\chi} + \sum_{p\neq q\neq o}\sum_{\chi,\mu,\xi}\sum_{n_1,n_2}-\frac{9i}{2}\omega_{pq}^{(n_1)}\omega_{qo}^{(n_2)}p_{\mu\chi\xi}^{(n_1,n_2)}\hat{I}_{p\mu}\hat{I}_{q\chi}\hat{I}_{o\xi}. \tag{31}$$

The second-order scaling factors $q_\chi^{(n_1,n_2)}$, $q_{\chi\mu}^{(n_1,n_2)}$, $p_\chi^{(n_1,n_2)}$ and $p_{\mu\chi\xi}^{(n_1,n_2)}$ for $\chi=x$ are given by

$$q_x^{(n_1,n_2)} = \sum_{\nu,\lambda}\frac{1}{\nu\omega_{\mathrm{m}}+\lambda\omega_{\mathrm{eff}}}\left(a_y^{(-\nu-c\cdot n_1,-\lambda)}a_z^{(\nu-c\cdot n_2,\lambda)} - a_z^{(-\nu-c\cdot n_1,-\lambda)}a_y^{(\nu-c\cdot n_2,\lambda)}\right), \tag{32}$$

$$\begin{aligned}q_{x,\mu}^{(n_1,n_2)} = \sum_{\nu,\lambda}\frac{1}{\nu\omega_{\mathrm{m}}+\lambda\omega_{\mathrm{eff}}}\Big(&a_y^{(-\nu-c\cdot n_1,-\lambda)}a_{z\mu}^{(\nu-c\cdot n_2,\lambda)} - a_z^{(-\nu-c\cdot n_1,-\lambda)}a_{y\mu}^{(\nu-c\cdot n_2,\lambda)}\\ &- a_y^{(\nu-c\cdot n_1,\lambda)}a_{z\mu}^{(-\nu-c\cdot n_2,-\lambda)} + a_z^{(\nu-c\cdot n_1,\lambda)}a_{y\mu}^{(-\nu-c\cdot n_2,-\lambda)}\Big),\end{aligned} \tag{33}$$

$$p_x^{(n_1,n_2)} = \sum_{\mu}\sum_{\nu,\lambda}\frac{1}{\nu\omega_{\mathrm{m}}+\lambda\omega_{\mathrm{eff}}}\left(a_{y\mu}^{(-\nu-c\cdot n_1,-\lambda)}a_{z\mu}^{(\nu-c\cdot n_2,\lambda)} - a_{z\mu}^{(-\nu-c\cdot n_1,-\lambda)}a_{y\mu}^{(\nu-c\cdot n_2,\lambda)}\right), \tag{34}$$

$$\begin{aligned}p_{\mu x\xi}^{(n_1,n_2)} = \sum_{\nu,\lambda}\frac{1}{\nu\omega_{\mathrm{m}}+\lambda\omega_{\mathrm{eff}}}\Big(&a_{\mu y}^{(-\nu-c\cdot n_1,-\lambda)}a_{z\xi}^{(\nu-c\cdot n_2,\lambda)} - a_{\mu y}^{(\nu-c\cdot n_1,\lambda)}a_{z\xi}^{(-\nu-c\cdot n_2,-\lambda)}\\ &- a_{\mu z}^{(-\nu-c\cdot n_1,-\lambda)}a_{y\xi}^{(\nu-c\cdot n_2,\lambda)} + a_{\mu z}^{(\nu-c\cdot n_1,\lambda)}a_{y\xi}^{(-\nu-c\cdot n_2,-\lambda)}\Big).\end{aligned} \tag{35}$$

Similar expressions result for $\chi=y$ and $z$ and can be found in the Supplementary Information (Section S1).

All first- and second-order scaling factors can be obtained from the Fourier coefficients $a_\chi^{(k,\ell)}$ characterizing the rf interaction-frame trajectory of the Cartesian spin operator $\hat{I}_z$ and thus do not depend on the details of the spin system. The effects of the additional rf-field modulations due to the radial contribution to the rf inhomogeneity will lead to changes in the interaction-frame trajectories and, therefore, changes in the scaling factors for the effective Hamiltonians compared to those calculated assuming a perfectly homogeneous rf field.

## 4 Methods and Materials

### 4.1 Numerical Simulations

The effect of the rf-field inhomogeneity on common solid-state NMR pulse sequences was investigated by numerical simulations in the usual rotating frame using the GAMMA spin-simulation environment (Smith et al., 1994). Unless otherwise noted,





spin dynamics were simulated at a $B_0$-field of 14.1 T corresponding to a proton resonance frequency of 600 MHz. Powder averaging was implemented according to the ZCW (Cheng et al., 1973) scheme using between 100 and 10000 crystallite orientations. A summary of the simulation parameters such as MAS frequencies and nominal rf field strengths for the individual pulse sequences that were investigated is given in the Supplementary Information (Table S1).

For all experimental schemes treated here, the general form of the time-dependent rf-field Hamiltonian is given by Eq. (19). Deviations from the nominal rf amplitude and phase due to the rf inhomogeneity are introduced by the relative amplitude and phase, denoted by $\omega_{1,\mathrm{rel}}(\boldsymbol{r},t)$ and $\phi_{\mathrm{rel}}(\boldsymbol{r},t)$ respectively. These parameters depend on the position of the crystallite in the sample space $\boldsymbol{r}$ and the rotor orientation. Under MAS, they are modulated by the rotor frequency. The time-dependent trajectories were computed numerically (s. Section 2) and are given as an input to the numerical simulations.

Simulations were performed for volume elements of the $rz$ plane indicated in yellow in Fig. 1a with an initial orientation of $\vartheta_0 = 0°$. For nutation experiments (s. Section 5.1), several $\vartheta_0$ values were considered, as the spin dynamics are in principle dependent on $\vartheta_0$ due to the non-commuting Hamiltonians at different time points during a rotor period. The sample space was restricted to the length of the coil along the rotor axis (indicated by dashed lines in Fig. 1b). Potential effects of the radial inhomogeneity should be similar for samples exceeding the length of the coil, but might be more pronounced since the magnitude of rf amplitude and phase modulations increases towards the rotor edges (s. Fig. 2). Simulation results of the individual volume elements were summed up during data processing and weighted with $r$ to account for the increase in volume with radial distance. The coil sensitivity (reciprocity theorem, Hoult and Richards (1976); Tošner et al. (2017)) was taken into account by additional weighting of each $rz$ element with the average relative rf amplitude over a rotor cycle $\bar{\omega}_{1,\mathrm{rel}}(\boldsymbol{r})$.

In order to separate the effect of the the static rf-field inhomogeneity from time-dependent effects due to amplitude and phase modulation arising from sample rotations, the spin dynamics were simulated under different conditions. Amplitude and phase modulations were considered separately and either treated as time dependent or as the static average over a rotor period. The four cases considered in this work are denoted as C1–C4, where

- C1: Time-averaged constant amplitude, zero phase
  $\bar{\omega}_{1,\mathrm{rel}}(\boldsymbol{r}) = \frac{1}{\tau_{\mathrm{r}}} \int_{t=0}^{\tau_{\mathrm{r}}} \omega_{1,\mathrm{rel}}(\boldsymbol{r},t)dt, \; \phi_{\mathrm{rel}}(\boldsymbol{r},t) = 0$

- C2: Time-dependent amplitude, zero phase:
  $\omega_{1,\mathrm{rel}}(\boldsymbol{r},t), \; \phi_{\mathrm{rel}}(\boldsymbol{r},t) = 0$

- C3: Time-averaged constant amplitude, time-dependent phase
  $\bar{\omega}_{1,\mathrm{rel}}(\boldsymbol{r}) = \frac{1}{\tau_{\mathrm{r}}} \int_{t=0}^{\tau_{\mathrm{r}}} \omega_{1,\mathrm{rel}}(\boldsymbol{r},t)dt, \; \phi_{\mathrm{rel}}(\boldsymbol{r},t)$

- C4: Time-dependent amplitude, time-dependent phase
  $\omega_{1,\mathrm{rel}}(\boldsymbol{r},t), \; \phi_{\mathrm{rel}}(\boldsymbol{r},t)$

Simulations with the time-averaged constant phase were not performed as constant phase offsets are small (maximum of less than $5°$) and the absolute phase of the rf irradiation has no influence on the outcome of an experiment. For reference, Table 1 summarizes the treatment of amplitude and phase modulations for these four cases.





**Table 1.** Summary of the treatment of the relative rf amplitude and phase for the four cases C1–C4.

|  | C1 | C2 | C3 | C4 |
|---|---|---|---|---|
| Amplitude | $\bar{\omega}_{1,\mathrm{rel}}(\boldsymbol{r})$ | $\omega_{1,\mathrm{rel}}(\boldsymbol{r},t)$ | $\bar{\omega}_{1,\mathrm{rel}}(\boldsymbol{r})$ | $\omega_{1,\mathrm{rel}}(\boldsymbol{r},t)$ |
| Phase | 0 | 0 | $\phi_{\mathrm{rel}}(\boldsymbol{r},t)$ | $\phi_{\mathrm{rel}}(\boldsymbol{r},t)$ |

## 4.2 Experimental

Experiments were performed on a 500 MHz Bruker Avance III HD NMR spectrometer equipped with a Bruker 1.9 mm triple-resonance MAS probe in double-resonance configuration at a temperature of 285 K. All powdered samples (natural-abundance glycine, natural-abundance L-histidine·HCl·H$_2$O) were purchased from commercial sources and used without further purification. Nutation spectra of glycine were recorded as two-dimensional experiments without sign discrimination in $t_1$ (simple sine amplitude modulation) at MAS frequencies of 15 and 30 kHz. The nominal rf-field amplitude was calibrated to 100 kHz using a nutation spectrum. Spectra were recorded with 512 $t_1$ increments with 12 scans each and a time increment for the nutation pulse of 2.5 µs. The spectral width in the direct dimension was set to 100 kHz and 1024 complex data points were acquired. Matlab (The MathWorks Inc., Natick, MA, U.S.A.) was used for data processing using a cosine-squared window function. Two-dimensional proton-proton correlation spectra of L-histidine with FSLG decoupling (Bielecki et al., 1989, 1990; Mote et al., 2016) in the indirect dimension were recorded at MAS frequencies of 14 and 28 kHz. Spectra were acquired with 512 $t_1$ increments with 8 scans each and time increments between 43.2 and 48 µs. States-type (States et al., 1982) data acquisition was used for phase sensitive detection and sign discrimination in $t_1$. The spectral width in the direct dimension was set to 200 kHz and 1024 complex data points were recorded. Nutation-frequency selective I-BURP-2 (Geen and Freeman, 1991) pulses in the spinlock frame were used for sample restriction (Aebischer et al., 2020). The rf-field amplitudes were calibrated using a nutation spectrum and set to 100 kHz during hard pulses and spinlock. The FSLG decoupling was implemented using shaped pulses with a time resolution of 100 ns for the phase ramp. Shape files with 80, 108 and 160 points were used, corresponding to nutation frequencies about the effective field of 250, 185.2 and 125 kHz. The carrier was placed outside the spectral region of interest, its position is indicated by an arrow. Spectra were processed in Matlab with zero-filling to 4096x4096 data points and the application of a cosine-squared apodization. The 1D spectra shown were obtained by summation over the relevant spectral region in $\omega_2$. Frequency axes in ppm were determined by comparison of the peak positions observed for the $\alpha$ and $\delta^2$ proton resonances in histidine with those found in the literature (Mithu et al., 2013).

## 5 Results and Discussion

In this section, we discuss how a number of common solid-state NMR experiments are affected by the MAS time-modulated radio-frequency fields. This is done by analytical calculations based on the Floquet description presented in Section 3.1, numerical simulations and in some examples using experimental data.



### 5.1 Nutation Spectroscopy

#### 5.1.1 Numerical Simulations and Experimental Results

Nutation spectra represent a simple method to characterize the rf-field distribution in the sample. Such spectra were simulated for one-spin systems and the rf inhomogeneity was included in the rf Hamiltonian of Eq. (19) setting the nominal phase of the rf-field to zero corresponding to rf irradiation along the $x$ axis. The nominal rf-field amplitude $\nu_{1,\text{nom}}$ was set to 100 kHz and the four cases C1–C4 (s. Table 1) were studied. As only isotropic spin interactions were considered, simulations were performed for a single crystallite orientation.

Simulated nutation spectra using the rf-field profiles of the 3.2 and 1.3 mm MAS probes at different spinning frequencies (15 kHz and 30 kHz, respectively) are shown in Fig. 3. The overall nutation profile of the 1.3 mm probe is narrower, indicating a more homogeneous rf-field distribution inside the coil and thus a less pronounced drop-off of the static rf-field amplitude along the rotor axis. Phase modulation of the rf field (C3 and C4) leads to sidebands at $0 \pm m \cdot \nu_{\text{r}}$ ($m = 1, 2$ are visible). The

275 intensity of these sidebands increases with increasing MAS frequency. Amplitude modulation of the applied rf field on the other hand (C2 and C4) leads to sidebands at $\nu_1 \pm m \cdot \nu_{\text{r}}$ (only $m = 1$ visible). These sidebands are significantly weaker than those arising due to phase modulations and their intensity increases with decreasing spinning frequency. The reduced intensity of these amplitude-modulation sidebands can be explained by the lower magnitude of amplitude modulations in comparison to phase modulations (see Fig. 2). Moreover, their position depends on the magnitude of the static rf-field amplitude and thus

varies depending on the position within the sample space. Both types of sidebands are weaker in the 1.3 mm rf profile indicating that rf amplitude and phase modulations are less pronounced in comparison to the 3.2 mm rf profile. The phases of the sidebands depend on the initial position $\vartheta_0$ of the simulated $rz$-plane (s. Fig. S1 in the Supplementary Information). However, the obtained spectra are very similar for all initial orientations and no significant influence of $\vartheta_0$ on the effects of the radial rf inhomogeneity has been observed.

Experimental [1]H nutation spectra of natural-abundance glycine measured at a proton resonance frequency of 500 MHz using a Bruker 1.9 mm MAS probe are shown in Fig. 4 for MAS frequencies of 30 kHz (Fig. 4a) and 15 kHz (Fig. 4b). As was observed in the simulated nutation spectra (s. Fig. 3), sidebands at $0 \pm m \cdot \nu_{\text{r}}$ due to rf phase modulations are visible in the experimental spectra. Moreover, sidebands at $\nu_{1,\text{nom}} \pm m \cdot \nu_{\text{r}}$ are visible that replicate the shape of the main nutation profile at

290 100 kHz. At lower MAS frequency, these sidebands increase in intensity whereas those at multiples of the rotor frequency are attenuated. In the simulated spectra shown in Fig. 3, the sidebands at 115 and 130 kHz were significantly weaker and did not have the same shape as the overall nutation profile. As is shown in Fig. S2 in the Supplementary Information, no such sidebands are observed in experimental nutation spectra of natural-abundance adamantane (see Fig. S2a in the SI) indicating that they arise from the MAS modulation of anisotropic interactions. This is confirmed by the simulated nutation spectra obtained for a

295 dipolar coupled two-spin system (see Fig. S2b in the SI), where strong side bands at $\nu_1 \pm \nu_r$ are obtained that nicely replicate the shape of the main nutation profile for all four cases C1–C4.





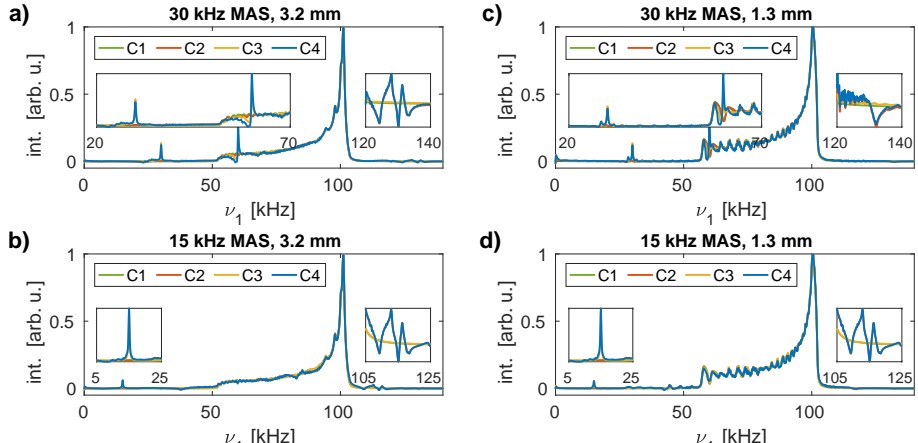

**Figure 3.** Simulated nutation spectra at a resonance frequency of 600 MHz for the 3.2 mm MAS probe (a and b) and the 1.3 mm MAS probe (c and d) for a nominal rf amplitude of 100 kHz. A spinning frequency of 30 kHz (15 kHz) was assumed for spectra in a and c (b and d). Sidebands at $0 \pm m \cdot \nu_r$ arise when rf phase modulations are taken into account (C3 and C4) while those at $\nu_1 \pm m \cdot \nu_r$ occur if amplitude modulations are present (C2 and C4). The amplitude-modulation sidebands increase in intensity at lower MAS frequencies, whereas the phase-modulation sidebands are attenuated. Both sideband families are less intense in the 1.3 mm probe and the overall nutation profile narrower, indicating an improved homogeneity of the rf distribution inside the coil.

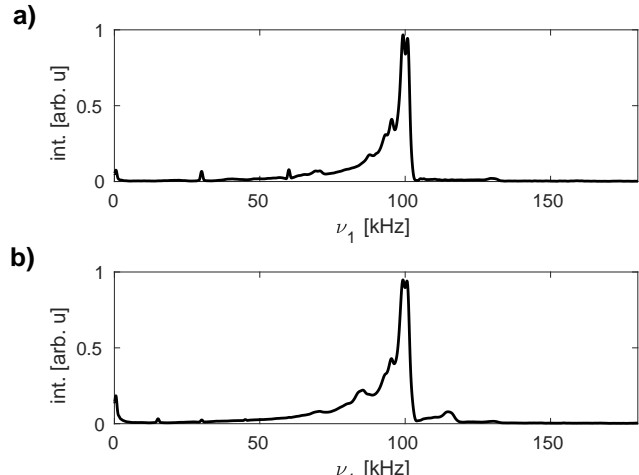

**Figure 4.** Experimental $^1$H nutation spectra of natural-abundance glycine recorded at a proton resonance frequency of 500 MHz in a Bruker 1.9 mm MAS probe spinning at 30 kHz (a) and 15 kHz (b). The nominal rf amplitude was set to 100 kHz as determined using a nutation spectrum. Sidebands due to rf phase modulations at $0 \pm m \cdot \nu_r$ for $m = 1, 2$ are visible in both spectra but have lower intensity at slower MAS as expected from the simulations (s. Fig. 3). Much broader sidebands that replicate the shape of the overall nutation profile are visible at $\nu_{1,\mathrm{nom}} \pm m \cdot \nu_r$ for $m = 1$. Their intensity increases significantly at the lower MAS frequency.





### 5.1.2 Floquet Analysis

Phase modulation of the rf field leads to non-commuting terms in the rf Hamiltonian at different points in time, thus prohibiting an analytical determination of the time evolution of the magnetization during the nutation experiment. However, insight can be gained from the interaction-frame trajectory of spin operators that can be computed numerically. For the $\hat{I}_z$ spin operator, this trajectory can be expanded as

$$\hat{\tilde{I}}_z(t) = a_x(t)\hat{I}_x + a_y(t)\hat{I}_y + a_z(t)\hat{I}_z \,. \tag{36}$$

Fourier analysis of the time-dependent $a_\chi(t)$ coefficients then yields the frequency components present in the nutation spectrum. Such interaction-frame trajectories of the $\hat{I}_z$ spin operator with rf irradiation along the $x$-axis were computed numerically in Matlab with a time resolution of 50 ns. A nominal rf-field strength of 100 kHz was chosen and a MAS frequency of 30 kHz assumed. The effects of MAS time-dependent rf amplitude and phase modulations were studied separately. Modulations were modeled as Fourier Series (see also Eqs. (1) and (2)) and magnitude $A_n^{(A/P)}$ and phase $\phi_n^{(A/P)}$ coefficients given as input.

Absolute values and phases of the $a_\chi(t)$ coefficients are shown in Figs. 5a and b as a function of the magnitude of amplitude modulations with $\omega_r$ and $2 \cdot \omega_r$ ($A_1^{(A)}$ and $A_2^{(A)}$). The static amplitude offset $A_0^{(A)}$ was set to 1 and all $A_n^{(A)}$ with $n \geq 3$ were set to zero. No phase modulation was taken into account. Amplitude modulation with the rotor frequency (Fig. 5a) leads to sidebands at $\nu_1 \pm m \cdot \nu_r$ with $m$ being any integer, whereas amplitude modulation with $2 \cdot \omega_r$ (Fig. 5b) leads to sidebands at $\nu_1 \pm 2m \cdot \nu_r$. The intensity of these sidebands increases with the magnitude of the modulation in both cases. However, sidebands arising from rf amplitude modulation with the base frequency $\omega_r$ are significantly stronger. The phase of the amplitude modulation $\phi_n^{(A)}$ for the spectra in Fig. 5 was set to zero as it only influences the phases of the centerband and sidebands. Static amplitude offsets $A_0^{(A)} \neq 1$ will simply shift the entire spectrum. Figures 5c and d show the absolute values and phases of $a_\chi(t)$ for phase modulations with $\omega_r$ and $2 \cdot \omega_r$ ($A_1^{(P)}$ and $A_2^{(P)}$). The static phase offset $A_0^{(P)}$ and all $A_n^{(A)}$ with $n \geq 3$ were set to zero. No rf amplitude modulations were taken into account. The frequency range shown in the figure was limited to 0–90 kHz since rf phase modulations lead to sidebands at $0 \pm n \cdot \nu_r$ that are significantly less intense than the main band at the nominal rf amplitude ($\nu_1 = 100$ kHz). In contrast to the sidebands observed for rf amplitude modulation, phase modulation with $n \cdot \omega_r$ exclusively leads to sidebands at $0 \pm n \cdot \nu_r$. Compared to the sidebands arising from amplitude modulation, the intensities of these phase modulation sidebands are considerably lower. Their intensity increases with the Fourier number $n$ of the modulation (side bands arising from $A_2^{(P)}$ are more intense than the ones from $A_1^{(P)}$). The phase of the modulation ($\phi_n^{(P)}$) was set to zero again, as it only affects the phase of the $a_\chi(t)$ coefficients (s. also Fig. S1 in the SI).

These results are in good agreement with the simulated and experimental nutation spectra shown in Figs. 3 and 4, where two separate families of sidebands arose for rf amplitude and phase modulation. As described earlier, the higher intensity observed for phase-modulation sidebands in these spectra can be explained by the overall larger magnitude of the phase modulations and the independence of the sideband position from the average rf-field amplitude. The position of the amplitude-modulation sidebands on the other hand shifts with the average static rf-field amplitude leading to a broadening of the sidebands.





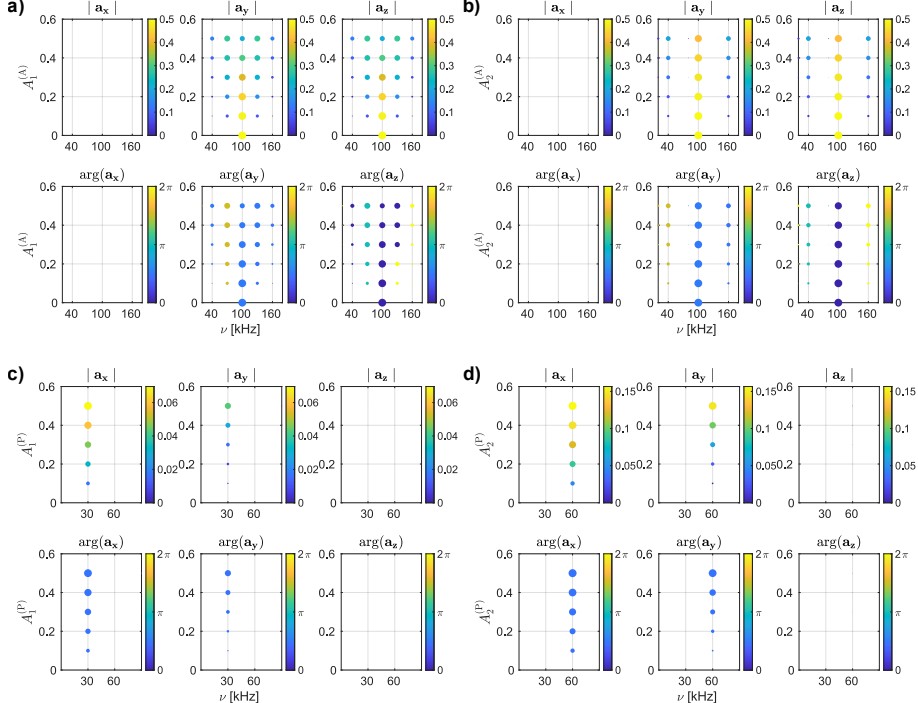

**Figure 5.** Absolute values and phases of the $a_\chi(t)$ coefficients characterizing the interaction-frame trajectory of the $\hat{I}_z$ operator during a nutation experiment with rf irradiation along the $x$-axis. A nominal rf-field strength of 100 kHz and a MAS frequency of 30 kHz was assumed. The magnitude and phase of the coefficients is shown as a function of the $A_n^{(\mathrm{A})}$ (a and b) and $A_n^{(\mathrm{P})}$ (c and d) magnitude coefficients for rf amplitude modulations with $\omega_\mathrm{r}$ (a) and $2 \cdot \omega_\mathrm{r}$ (b) as well as phase modulations with $\omega_\mathrm{r}$ (c) and $2 \cdot \omega_\mathrm{r}$ (d). Sidebands due to the MAS modulation of the rf amplitude at $\nu_1 \pm m \cdot \nu_\mathrm{r}$ with $m$ being any integer for modulations with $\omega_\mathrm{r}$ ($A_1^{(\mathrm{A})}$) and any even integer for modulations with $2 \cdot \omega_\mathrm{r}$ ($A_2^{(\mathrm{A})}$) are observed. The intensity of these sidebands increases with the strength of the modulation. For rf phase modulations, sidebands at $0 \pm n \cdot \nu_\mathrm{r}$ arise for modulations with $n \cdot \omega_\mathrm{r}$. The intensity of these sidebands increases with the strength of the modulation and considerably higher intensities are observed for higher $n$ (note the different scaling of the colourbars for $A_1^{(\mathrm{P})}$ and $A_2^{(\mathrm{P})}$ in c and d). The frequency axes in c) and d) was restricted to 0–90 kHz since the phase modulation sidebands are significantly weaker than the main band contribution to $a_z(t)$ at the nominal rf amplitude of 100 kHz.

## 5.2 Cross-Polarization

Hartmann-Hahn cross polarization (Hartmann and Hahn, 1962; Stejskal et al., 1977) is probably the most ubiquitous pulse-sequence element in solid-state NMR. Under MAS, the sum or difference of the two rf-field amplitudes have to be matched to an integer multiple of the spinning frequency

$$\omega_{1\mathrm{S}} \pm \omega_{1\mathrm{I}} = n \cdot \omega_\mathrm{r} \quad n = \pm 1, \pm 2 \tag{37}$$

Due to the difference of the rf-field inhomogeneity of the two rf fields across the sample, this condition cannot be fulfilled simultaneously in the entire sample volume and only certain parts of the sample will participate in the polarization transfer





thus decreasing the resulting signal intensity. One popular strategy to overcome this volume selectivity is ramped-amplitude cross polarization (Metz et al., 1994) or the adiabatic modulation of the rf-field amplitude during the contact time (Hediger
et al., 1995) (s. Supplementary Information Section S4 for more details). In this work, $\hat{I}_x \rightarrow \hat{S}_x$ magnetization transfers at the $n = 1$ zero-quantum matching conditions in heteronuclear CN, HN and HC two-spin systems were simulated for standard, ramped-amplitude and adiabatic-passage CP experiments for the 3.2 mm MAS probe at a proton resonance frequency of 600 MHz. A MAS frequency of 20 kHz was assumed and the nominal rf fields and contact times set to CN: 85 kHz on C, 65 kHz on N, 5 ms contact time; HN: 70 kHz on H, 50 kHz on N, 1 ms contact time and HC: 90 kHz on H, 70 kHz on C, 1
345 ms contact time. The anisotropy of the dipolar-coupling tensor $\delta_{IS} = -2\frac{\mu_0 \hbar \gamma_I \gamma_S}{4\pi r_{IS}^3}$ was estimated from average bond lengths and $\frac{\delta_{IS}}{2\pi}$ values of 1.9 kHz, 25 kHz and -46 kHz were used for the CN, HN and HC simulations respectively. Chemical shifts as well as $J$-coupling constants were set to zero. The simulations were performed with a time resolution of 250 ns and the $x$-magnetization of both the source and the destination spin detected every 5 μs. Powder averaging was performed over 1154 crystallite orientations.

The simulated time evolution of the spin-locked $x$-magnetization on both spins for the cases C1–C4 (s. Table 1) is shown in Fig. 6 for all three spin pairs. The rf amplitude on the source spin was either kept constant (left-hand column), modulated with a linear ramp (middle column) or using a tangential modulation (right-hand column) (s. Fig. S3 in the SI for more details). In CN spin pairs, a tangential modulation of the rf amplitude on one of the spins leads to a significant improvement of the transfer
efficiency (up to 35 %) compared to both the standard and the ramped amplitude CP experiment (less than 20 %). For spin pairs with stronger dipolar couplings such as HN and HC, both the ramped amplitude CP as well as the adiabatic passage CP lead to similar transfer efficiencies of up to 70 %. In all simulated experiments, only marginal differences between the obtained transfer efficiencies for the four cases C1–C4 are observed for all spin pairs. Time-dependent modulations of the rf amplitude and phase due to the radial rf inhomogeneity, therefore, do not seem to have a significant effect on the magnetization build up
on the destination spin. On the source spin, some magnetization is lost when rf phase modulations are present (C3 and C4) which is also observed for one-spin spin-lock simulations. As the radial contributions to the rf inhomogeneity are weaker in the 1.9 mm and 1.3 mm probes, similar results would be expected for these probes. Overall, these simulation results suggest, that the effect of the radial inhomogeneity on CP polarization transfers is negligible. Only the static rf amplitude offset over the relevant sample space is important due to the volume selectivity it causes.
Moreover, polarization transfers in NCA and NCO two-spin systems using the tm-SPICE sequences (Tošner et al., 2018) were simulated. These pulse schemes were developed using optimal-control (OC) strategies taking into account the MAS modulations of the rf field due to the radial rf inhomogeneity. The resulting magnetization transfers are shown in Fig. 7 for the 3.2 mm MAS probe at a proton resonance frequency of 400 MHz. Nominal rf amplitudes on both channels were set to 40 kHz and a spinning speed of 20 kHz assumed. The shape files for the pulse sequences contain 1750 points with a time resolution of
2 μs, corresponding to a contact time of 3.5 ms. The time resolution for the propagation was set to 250 ns and the expectation value of the $\hat{I}_x$ and $\hat{S}_x$ operators detected every 5 μs. 1154 crystallite orientations were used for the powder averaging. Spin system parameters ($J$-couplings, CSA and dipolar coupling tensors) were taken from Tošner et al. (2018) and can be found in





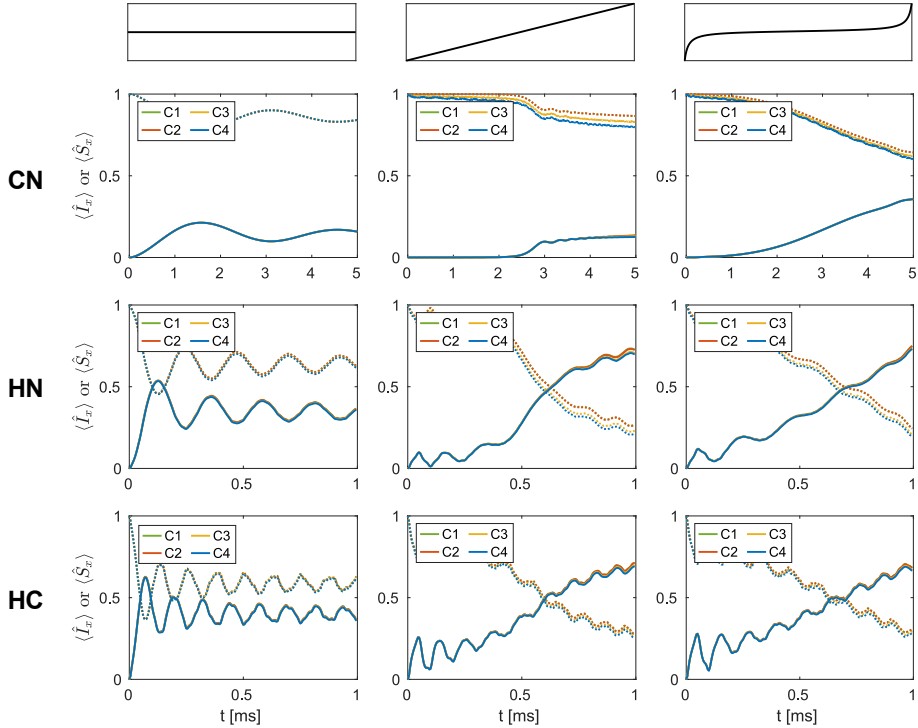

**Figure 6.** Simulated time evolution of the expectation value of the $\hat{I}_x$ (dotted lines) and $\hat{S}_x$ (solid lines) operators during CP zero-quantum $\hat{I}_x \rightarrow \hat{S}_x$ polarization transfers in CN (top row), HN (middle row) and HC (bottom row) spin pairs for C1–C4 (s. Table 1) in a 3.2 mm probe at a proton resonance frequency of 600 MHz assuming a MAS frequency of 20 kHz. Shown are magnetization transfers for constant rf amplitude (left-hand column), a linear rf amplitude modulation (middle column) and a tangential rf amplitude modulation on the source spin (right-hand column). The rf amplitude on the destination spin was kept constant (65 kHz on N in CN, 50 kHz on N in HN and 70 kHz on C in HC), modulations of the rf amplitude on the source spin are shown in Fig. S3 in the Supplementary Information. Overall, no significant effects of the time-dependent amplitude and phase modulations due to radial contributions to the rf inhomogeneity are observed for the destination spin in any of the simulated transfers.

Tables S2 and S3 in the Supplementary Information. Impressive transfer efficiencies of around 60% are obtained for both NCA (Fig. 7a) and NCO (Fig. 7b) spin pairs. However, only minor differences between the four cases C1–C4 are observed in these
simulations. This reflects the fact that the optimization of this sequence took rf amplitude and phase modulations of different magnitude into account as well as varying initial phases of these modulations. Therefore, the sequence performs well under all possible conditions encountered in the rotor leading to an increase in the robustness of the resulting sequences towards static as well as time-dependent rf inhomogeneity. The broad range of considered conditions stabilizes the optimization towards a broader minimum that gives better transfer over the complete rotor.





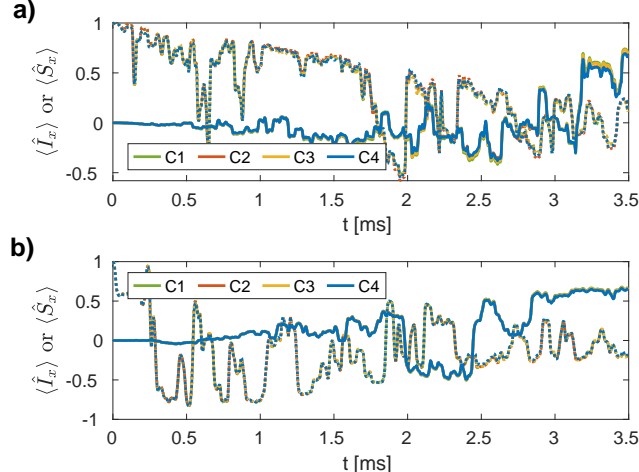

**Figure 7.** Simulated time evolution of the $\hat{I}_x$ (dotted lines, $^{15}$N) and $\hat{S}_x$ (solid lines, $^{13}$C) operators during tm-SPICE (Tošner et al., 2018) CP polarization transfers in NCA (a) and NCO (b) two-spin systems (s. Tables S2 and S3 in the SI for spin system parameters) in a 3.2 mm MAS probe at a proton resonance frequency of 400 MHz. A spinning frequency of 20 kHz was assumed and nominal rf amplitudes set to 40 kHz on both channels. Impressive transfer efficiencies of approximately 60% are achieved in both spin pairs. However, no significant differences between the four cases C1–C4 are observed.

## 5.3 Rotational-Echo Double Resonance

In Rotational-Echo Double Resonance Recoupling (REDOR) (Gullion and Schaefer, 1989a, b) the heteronuclear dipolar coupling is reintroduced by trains of two rotor-synchronized $\pi$ pulses per rotor cycle. This technique allows the quantitative measurement of dipolar couplings in heteronuclear spin pairs and has become a valuable tool in the characterization of structure (Hong, 2006; Rienstra et al., 2002; Michal and Jelinski, 1997; Jia et al., 2015) and dynamics (Schanda et al., 2010).

Numerical simulations of REDOR recoupling were performed for CN and HN spin pairs at a proton resonance frequency of 600 MHz with an XY-4 phase cycling scheme (Gullion et al., 1990) as it is commonly implemented to generate a pure Ising-type Hamiltonian. Dipolar couplings were estimated from average bond lengths and the anisotropy of the coupling $\frac{\delta_{IS}}{2\pi}$ set to to 2 kHz and 24 kHz in CN and HN, respectively. Chemical-shift tensors and scalar $J$-couplings were set to zero. Resulting simulated REDOR curves for CN spin pairs are shown in Fig. 8 for a 1.3 mm (Fig. 8a) and a 3.2 mm (Fig. 8b) probe assuming a spinning frequency of 20 kHz. The nominal rf field strengths were set to 100 kHz (62.5 kHz) on C and 65 kHz (50 kHz) on N in the 1.3 mm (3.2 mm) probe. A time resolution of 250 ns was chosen for the propagation and 538 crystallite orientations used for the powder averaging. Compared to the theoretical REDOR curve (dashed line, analytical expression including finite-pulse effects can be found in Jaroniec et al. (2000)), considerably lower recoupling efficiencies are obtained for C1–C4 in both probes (Nishimura et al., 2001). However, only minor differences between the four cases are observed with amplitude modulations (C2 and C4) leading to a slight deterioration of the recoupling performance. These effects are very





similar for both probes. The relative timing of the rotor-synchronized $\pi$ pulses in the REDOR sequence with respect to the time-dependent rf-field amplitude and phase modulations only has a marginal effect on the recoupling performance (s. Fig. S5 in the Supplementary Information for further details). Simulated REDOR curves for the HN spin system in the 1.9 mm probe assuming a spinning frequency of 40 kHz are shown in Figs. 8c and d. As the dipolar coupling in the HN spin pair is too large to allow sufficient sampling of the REDOR curve, modified REDOR implementations were simulated in which one (Gullion and Schaefer, 1989b) (Fig. 8c) or both (Jain et al., 2019) (Fig. 8d) of the pulses in the basic building block are shifted. The corresponding pulse sequences are shown in Fig. S4 in the SI. These schemes lead to a scaling of the effective dipolar coupling and thus allow sufficient sampling of the REDOR curve even for strongly dipolar-coupled spin pairs. A time resolution of 125 ns was chosen and the nominal rf fields set to 125 kHz on H and 50 kHz on N. Powder averaging was performed for 10000 crystallite orientations. Theoretical REDOR curves are again indicated by the dashed lines (analytical expressions including finite-pulse effects can be found in Schanda et al. (2011) for Fig. 8c and Jain et al. (2019) for Fig. 8d). For both shifting regimes only slight deviations from the theoretical curves are observed. Moreover, resulting REDOR curves for C1–C4 are identical, indicating that time-dependent amplitude and phase modulations have no effect on the recoupling performance in strongly dipolar coupled spin pairs. Overall, the REDOR sequence seems to be predominantly affected by the static rf inhomogeneity which causes deviations of the pulse flip angles from the desired $180°$ due to average rf-field amplitude deviations.

### 5.4 Symmetry-Based C$N$ Recoupling - C7 and POST-C7

Symmetry-based $\mathrm{C}N_\kappa^\nu$ sequences represent an important class of homonuclear recoupling sequences. Since the first introduction of the original $\mathrm{C}7_2^1$ sequence (Lee et al., 1995) many other symmetry-based sequences have been proposed and characterized, however only the $\mathrm{C}7_2^1$ and the POST-C7 sequence (Hohwy et al., 1998), where the basic $\mathrm{C}_\phi = (2\pi)_\phi (2\pi)_{\phi+\pi}$ is replaced by the cyclically permuted $\mathrm{C}_\phi = (\frac{\pi}{2})_\phi (2\pi)_{\phi+\pi} (\frac{3\pi}{2})_\phi$ POST element, were considered in this work.

Numerical simulations of $\hat{S}_{1z} \rightarrow \hat{S}_{2z}$ polarization transfers during C7 and POST-C7 recoupling were performed for CC two-spin systems under conditions typical for a 3.2 mm MAS probe at a carbon resonance frequency of 150 MHz. The nominal rf-field amplitude was set to 70 kHz and a spinning frequency of 10 kHz assumed. A time resolution of approximately 714 ns was chosen and 538 crystallite orientations used for the powder averaging. The time evolution of the expectation values of $\hat{S}_{1z}$ and $\hat{S}_{2z}$ in a CC spin pair with isotropic chemical shifts that are symmetric around zero ($\Omega_1 = -\Omega_2$) is shown in Fig. 9 for C7 (Fig. 9a) and POST-C7 (Fig. 9b) for the cases C1–C4. The anisotropy of the dipolar coupling tensor was estimated from average bond lengths and $\frac{\delta_{IS}}{2\pi}$ set to 4.5 kHz. For both sequences, transfer efficiencies of approximately 70% are achieved in the first transient for a mixing time of approximately 10 ms. Amplitude modulations due to the radial rf-field inhomogeneity (C2 and C4) lead to a slight deterioration of the recoupling performance. This effect is more pronounced for C7, indicating the improved robustness of the POST-C7 sequence. At longer times, a loss of magnetization on both spins is observed when time-dependent amplitude modulations are taken into account. Interestingly, the overall order of the cases is different for the two sequences. For C7 recoupling, higher transfer efficiencies are observed for C4 in comparison to C2, whereas this order is reversed for POST-C7. Thus POST-C7 seems to be more sensitive to combined amplitude and phase modulations (C4). Figures

MAGNETIC
RESONANCE
Open Access Discussions

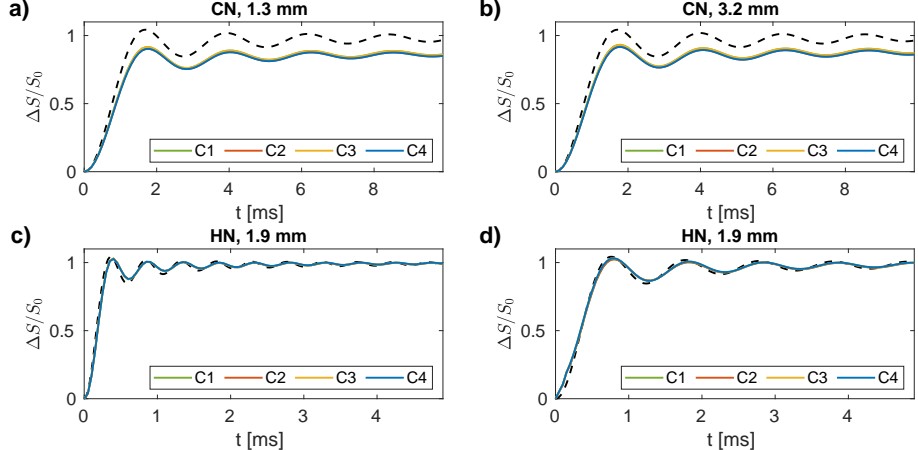

**Figure 8.** a) / b) Simulated REDOR curves for a CN spin pair with a value of 2 kHz for the anisotropy of the dipolar coupling tensor $\frac{\delta_{IS}}{2\pi}$ at a carbon resonance frequency of 150 MHz in a 1.3 mm MAS probe (a) and a 3.2 mm MAS probe (b) assuming a spinning frequency of 20 kHz. The nominal rf-field strengths were set to 100 kHz (62.5 kHz) for C and 62.5 kHz (50 kHz) for N in the 1.3 mm (3.2 mm) probe. For all four cases C1–C4, the resulting recoupling efficiencies are significantly lower than the theoretical REDOR curve (dashed line). Amplitude modulations (C2 and C4) lead to a further marginal deterioration of the recoupling efficiency. The remaining two cases (C1 and C3) are indistinguishable. c) / d) Simulated REDOR curves for a HN spin pair with a $\frac{\delta_{IS}}{2\pi}$ of 24 kHz at a proton resonance frequency of 600 MHz in a 1.9 mm MAS probe assuming a spinning frequency of 40 kHz. The nominal rf-field strengths were set to 125 kHz for H and 50 kHz for N. A scaling of the effective dipolar coupling is achieved by shifting one pulse (c, delay until first pulse $t_1 = 2.5$ μs) or both pulses (d, delay until first pulse $t_1 = 16$ μs) per rotor period. No significant deviation of any of the four cases C1–C4 from the theoretical REDOR curves (dashed lines) is observed indicating the robustness of these REDOR implementations towards rf inhomogeneity.

9c (C7) and d (POST-C7) show simulation results of the same spin system for a spatially restricted sample (central third along the rotor axis). This restriction of the sample space mitigates the effects of amplitude modulations and only very marginal differences between the four cases are observed for both sequences. In order to further investigate the robustness of the two sequences, simulations in a second model system with a large CSA tensor were performed at a lower external magnetic field

(75 MHz carbon resonance frequency). The parameters of this spin system were based on phthalic acid (Hellwagner et al. (2017), s. Table S4 in the SI for details). Significantly lower overall transfer efficiencies (approximately 50%) are observed (s. Fig. 9c and d) and for both sequences rf-field amplitude modulations (C2 and C4) further deteriorate the recoupling efficiency. This decrease is less pronounced for POST-C7 (s. Fig. 9d), again indicating its improved robustness. We have not carried out a complete Floquet analysis of the effects caused by the time-dependent rf-field amplitudes. We believe that the decreased

efficiency is due to the appearance of effective fields that shift the resonance condition slightly as is the case for pulse transients (Hellwagner et al., 2017) .



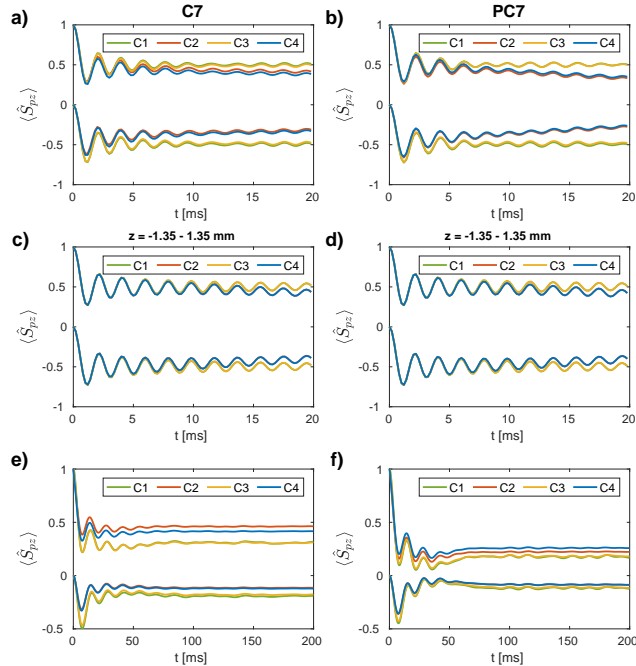

**Figure 9.** Simulated $\hat{S}_{1z} \rightarrow \hat{S}_{2z}$ polarization transfers in CC spin systems during C7 (a, c, e) and POST-C7 (b, d, f) recoupling for C1–C4 in a 3.2 mm probe at a carbon resonance frequency of 150 MHz. A nominal rf-field amplitude of 70 kHz and a spinning frequency of 10 kHz was assumed. Resulting expectation values of $\hat{S}_{1z}$ and $\hat{S}_{2z}$ for a spin pair with isotropic chemical shifts that are symmetric around zero ($\Omega_1 = -\Omega_2$) and a value of 4.5 kHz for $\frac{\delta_{IS}}{2\pi}$ are shown for the full rotor (a and b) and for a spatially restricted sample (central third along rotor axis, c and d). Overall, rf-field amplitude modulations (C2 and C4) lead to a slight deterioration of the recoupling performance for both sequences. The other two cases (C1 and C3) are nearly identical. e) / f) Simulation results for a CC spin pair with considerable CSA and a value of -585 Hz for $\frac{\delta_{IS}}{2\pi}$ at a lower magnetic field (carbon resonance frequency of 75 MHz) for C7 (e) and POST-C7 (f) recoupling. Lower overall transfer efficiencies are observed for this spin system and modulations of the rf-field amplitude (C2 and C4) further deteriorate the recoupling performance (C1 and C3 are indistinguishable). This effect is stronger for C7.

## 5.5 Frequency-Switched Lee–Goldburg Decoupling

### 5.5.1 Numerical Simulation and Experimental Results

Frequency-Switched Lee–Goldburg (FSLG) decoupling is a homonuclear dipolar decoupling technique that can be used in
combination with MAS to improve resolution of spectra for dipolar-coupled homonuclear spin systems (Lee and Goldburg, 1965; Goldburg and Lee, 1963; Bielecki et al., 1989, 1990). The experiment is based on off-resonance rf irradiation leading to a truncation of the second-rank spin tensor of the homonuclear dipolar coupling by an effective radio-frequency field inclined at an angle $\theta_m \approx 54.74°$ with respect to the static magnetic field. Experimentally, FSLG can also be implemented using on-resonance irradiation with a constant rf-field amplitude and a continuous phase ramp to generate the frequency offset. The total
cycle time is divided into two intervals of equal length during which the phase is rotated in opposite directions (inverting the





offset) and with a phase jump of $180°$ in between.

The effects of the radial part of the rf-field inhomogeneity on the residual line width under FSLG decoupling were simulated for a homonuclear dipolar-coupled three-spin system in a 3.2 mm MAS probe at a proton resonance frequency of 600 MHz assuming a MAS frequency of 12.5 kHz. The nominal rf-field amplitude was set to 102.06 kHz corresponding to a tilting along the magic angle of an effective field with a strength of 125 kHz (FSLG cycle time of 16 μs). Using such a synchronization between the FSLG sequence and the sample spinning makes the simulations much more efficient than an asynchronous implementation while at the same time avoiding all resonance conditions up to and including second order. The FSLG decoupling was implemented using a phase ramp with a time resolution of 50 ns. The same time resolution was chosen for the propagation of the Hamiltonian. The initial density operator was set to $\hat{F}_y = \sum_{p=1}^{3} \hat{I}_{py}$ and transverse magnetization components detected every 48 μs (three FSLG cycles). A total of 8192 data points were acquired and the FID processed in Matlab. Powder averaging was performed over 1154 orientations. The parameters characterizing the chemical-shift and dipolar-coupling tensors were chosen to mimic a $CH_2$ group with couplings to an additional remote spin and can be found in the SI in Tables S5 and S6. Scalar $J$-couplings were neglected and set to zero.

Simulated spectra of the three-spin system are shown in Fig. 10a for C1–C4. In all four cases, all resonances show strong asymmetric features on the left-hand side of the spectral line due to the distribution of the chemical-shift scaling factors (Hellwagner et al., 2020). Considerable additional line broadening is observed when amplitude modulations are taken into account (C2 and C4). As the same linewidths were obtained in simulations of an asynchronous implementation of FSLG decoupling (MAS frequency of approximately 14.1 kHz, s. Fig. S6 in the Supplementary Information), the broadening is not caused by resonance effects. This effect is observed for all three resonances but is most pronounced for the $CH_2$ resonance around 1.25 kHz. The additional time dependence of the rf phase in C4 results in no additional broadening and the two remaining cases (C1 and C3) are indistinguishable. Phase modulation, therefore, does not seem to have an influence on the obtained line width. Figure 10 also shows simulated FSLG spectra for radial slices of the simulated $rz$-plane at $r = 0.65$ (Fig. 10b) and 1.3 mm (Fig. 10c) and for a spatially restricted sample space (central third along the rotor axis, $z = -1.35 - 1.35$ mm, all $r$ values, Fig. 10d). As the magnitude of the rf amplitude modulations increases towards the rotor edges (s. Fig. 2), significantly stronger broadening is observed for radial slices closer to the coil windings. Spatial restriction of the sample space to the central third leads to a reduction of the line width. This line narrowing is significantly more pronounced for the resonance at -2.75 kHz where the foot on the left-hand side of the resonance is eliminated for all four cases. For C2 and C4, broadening is observed even in this spatially restricted sample, indicating that time-dependent amplitude modulations without a static rf amplitude offset still result in contributions to the residual line width. Simulations were also performed for a six spin system and qualitatively similar results were obtained.

In order to observe this broadening experimentally, the sample space has to be restricted to areas close to the coil windings where strong rf-field amplitude modulations occur. This could in principle be achieved by physically restricting the sample using cylindrical spacers. However, homogeneous packing in such a sample is difficult to achieve. Alternatively nutation-





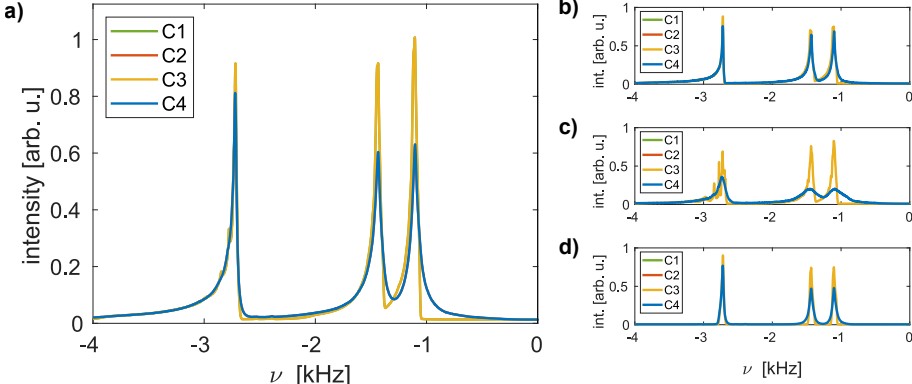

**Figure 10.** Simulated FSLG decoupled proton spectra of a three-spin system at a resonance frequency of 600 MHz for a 3.2 mm MAS probe assuming a MAS frequency of 12.5 kHz for C1–C4. The rf-field strength along the magic angle was set to 125 kHz. a) Spectrum for the full $rz$-plane. Significant additional line broadening is observed when amplitude modulations are taken into account (C2 and C4). The two remaining cases are indistinguishable, indicating that rf phase modulations do not contribute to the residual line width. b) / c) / d) Contributions from radial slices (all $z$ values for given $r$) at $r = 0.65$ (b) and 1.3 mm (c) as well as the spectrum of a spatially restricted sample (central third along the rotor axis, $z = -1.35 – 1.35$ mm, all $r$ values, d). Stronger line broadening is observed towards the edges of the rotor where rf modulations are more pronounced. Significantly narrower lines are obtained for the restricted sample, however, considerable broadening is still observed when time-dependent rf amplitude modulations are taken into account (C2 and C4). The observed splitting of the line for C1 and C3 in d) can attributed to the distribution of the isotropic chemical-shift scaling factors in this radial slice (s. Fig. S7 in the SI for details).

frequency selective pulses as described in Aebischer et al. (2020) can be used to select the desired areas which also correspond to high average rf-field amplitudes. Figure 11 shows FSLG decoupled proton spectra of L-histidine measured at a proton resonance frequency of 500 MHz in a Bruker 1.9 mm MAS probe using a 2 ms I-BURP-2 pulse in the spinlock frame for the $B_1$-field selection of areas where the rf-field amplitude corresponds to the nominal value (s. Fig. S8 in the SI for a simulated inversion profile). Spectra were recorded with different $B_1$-field strengths for the FSLG decoupling at spinning frequencies of 14 and 28 kHz. No significant improvement of the obtained line width is observed for higher MAS frequencies and stronger decoupling field strengths. This indicates that the residual line width in these spectra is not decoupling limited, therefore, prohibiting the experimental characterization of the additional broadening caused by rf-field amplitude modulations due to the radial rf-field inhomogeneity.

### 5.5.2 Floquet Analysis

In order to gain physical insight into the origin of the observed line broadening in FSLG-decoupled spectra due to rf-field amplitude modulations, scaling factors for the first and second-order contributions to the effective Hamiltonian were computed (s. Section 3.1.1 for more details). As a simple measure for the magnitude of contributing first-order terms the norms of



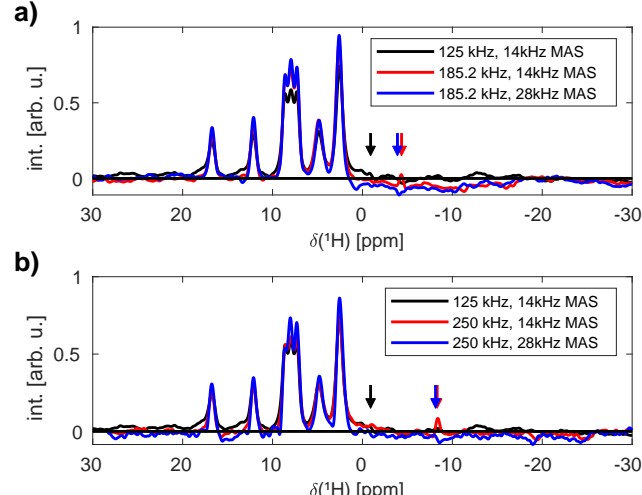

**Figure 11.** Experimental FSLG decoupled proton spectra of natural-abundance L-histidine recorded at a proton resonance frequency of 500 MHz using a 1.9 mm Bruker MAS probe. All spectra were recorded using a 2 ms I-BURP-2 pulse with a modulation frequency of 100 kHz (corresponding to the nominal rf-field amplitude as determined using a nutation spectrum) for the $B_1$-selection (Aebischer et al., 2020). The effective rf-field strength along the magic angle during the FSLG decoupling was set to 125 (a and b), 185.2 (a) and 250 kHz (b) at MAS frequencies of 14 and 28 kHz. No significant improvements are observed for stronger $B_1$-fields and higher spinning frequencies, indicating that the residual linewidth is not decoupling limited.

one-spin

$$\bar{a}_\chi^{(k)} = \sqrt{\sum_{\chi'} \left| \sum_{\ell=-1}^{1} a_{\chi'}^{(k,\ell)} \right|^2} \qquad (38)$$

and two-spin coefficients

$$\bar{a}_{\chi\mu}^{(k)} = \sqrt{\sum_{\chi',\mu'} \left| \sum_{\ell=-2}^{2} a_{\chi'\mu'}^{(k,\ell)} \right|^2} \qquad (39)$$

were computed.

Interaction-frame trajectories using the rf-field distribution in a 3.2 mm MAS probe during FSLG decoupling were computed numerically in Matlab with a time resolution of 50 ns and the Fourier coefficients extracted. The spinning frequency was chosen to be 12.5 kHz and the effective field strength along the magic angle set to 125 kHz, corresponding to a modulation frequency of the rf Hamiltonian of 62.5 kHz. This leads to the synchronization of the MAS rotation and the rf irradiation after a single rotor cycle or five FSLG cycles. This choice of frequencies should avoid all resonance conditions up to and including second order. Relative rf-field amplitude and phase modulations were modeled as Fourier series and fitted Fourier coefficients





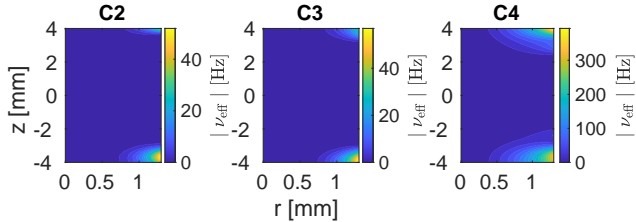

**Figure 12.** Effective nutation frequencies $\nu_{\text{eff}}$ over a MAS period during FSLG decoupling as a function of the position within the 3.2 mm MAS probe for C2, C3 and C4. Effective fields were extracted from interaction-frame trajectories computed for a field strength of 125 kHz along the magic angle and a MAS frequency of 12.5 kHz. Small effective fields of up to 50 Hz arise for rf amplitude and phase modulations alone (C2 and C3). Substantially larger fields up to 400 Hz are observed for combined modulations (C4). In general, effective fields of considerable size are only obtained at the rotor edges and even the maximum resulting magnitudes remain small compared to the nominal rf-field strength and the rotor frequency.

up to fourth order used as input (see Section 2). In analogy to the treatment of the rf-field inhomogeneity in numerical simulations, amplitude and phase modulations were considered separately and the four cases C1–C4 summarized in Table 1 studied.

A full FSLG cycle assuming a time-independent rf Hamiltonian and an ideal phase ramp with a $180°$ phase shift in the middle consists of two $\beta$ rotations with opposite direction about the effective field. The overall propagator would thus be the unity operator and $\omega_{\text{eff}} = 0$ in Eq. (12). However, the time-dependent modulations of the rf amplitude and phase due to the radial rf inhomogeneity can give rise to an additional effective field. The magnitude of this field as a function of the position within the sample space in the 3.2 mm probe is shown in Fig. 12. As the static rf-field inhomogeneity does not lead to additional effective fields, only cases where either the rf-field amplitude, the phase or both are time-dependent (C2, C3 and C4) are shown. Effective fields arise mainly at the edges of the rotor (large $r$ and $z$) where modulations are strongest. Amplitude and phase modulations alone (C2 and C3 respectively) lead to very small effective fields (max. 50 Hz) whereas larger effective fields (up to 400 Hz) result for combined amplitude and phase modulations (C4). In comparison to the rotor frequency and the basic modulation frequency of the FSLG sequence, these additional fields are small and will most likely not have any significant effects except for a small change in the effective field direction and magnitude.

To first order, the relevant scaling factors are those of the chemical shift ($a_\chi^{(k,\ell)}$ with $k = 0, \pm 1, \pm 2$) and those of the dipolar coupling ($a_{\chi\mu}^{(k,\ell)}$ with $k = \pm 1, \pm 2$). These can contribute to the first-order effective Hamiltonian (see Eq. (27)) since the modulation by the rf-field amplitude can be compensated by the time dependence due to MAS. The resulting norm of the $a_\chi^{(k,\ell)}$ coefficients (Eq. (38)) as a function of the position within the sample space is shown in Fig. 13a for C1–C4. As coefficients are symmetric ($\bar{a}_\chi^{(k)} = \bar{a}_\chi^{(-k)}$), only those corresponding to $k = 0, 1$ and 2 are shown. The scaling of the isotropic chemical shift ($k = 0$) is close to the ideal value of $\cos(\theta_{\text{m}}) \approx 0.577$ in regions of the rotor where the rf-field amplitude is comparable to the nominal value. Towards the edges of the rotor, the rf-field amplitude decreases, leading to a smaller tilt angle of the effective



field during FSLG and thus an increase of the scaling factor. The time-modulated part of the rf-field inhomogeneity does not appear to have any influence on the isotropic chemical shift, as no significant differences between the four cases are observed.

Time-dependent amplitude modulations (C2 and C4) lead to additional non-zero coefficients for chemical-shift contributions with $k \neq 0$ that can also contribute to the first-order effective Hamiltonian for $k = \pm 1, \pm 2$ (see Eq. (27)) where parts of the CSA tensor become time-independent. These contributions will be strongest at the very edges of the sample space (large $r$ and $z$) but non-zero coefficients are also obtained in the central third of the rotor close to the coil windings.

Under ideal conditions, the FSLG decoupling scheme leads to the averaging of the anisotropic dipolar coupling to first order and the corresponding scaling factors would be zero. However, dipolar-coupling terms are reintroduced when rf modulations are taken into account. The norm of the relevant $a_{\chi\mu}^{(k,\ell)}$ coefficients (Eq. (39)) is shown in Fig. 13b for the 3.2 mm MAS probe. Again, only the $k = \pm 1, \pm 2$ terms can contribute to the first-order effective Hamiltonian and partially reintroduce Fourier components of the dipolar coupling. As was the case for the chemical-shift scaling factors, the coefficients are symmetric and thus

only those for $k = 1$ and $2$ are shown. Amplitude modulations (C2 and C4) lead to significant $k = 1$ scaling factors and non-zero coefficients are not only obtained at the very edges of the rotor but also in the central third close to the coil windings. The additional phase modulation in C4 does not have an influence and amplitude modulations alone thus seem to be responsible for the reintroduction of the first-order coupling terms. The contribution of individual $a_{\chi\mu}^{(k=1)}$ coefficients to the norm are shown in Fig. S9 in the Supplementary Information for C4. Significant $a_{\chi\mu}^{(k=1)}$ are obtained for $\hat{I}_{pz}\hat{I}_{qx}$, $\hat{I}_{px}\hat{I}_{qx}$ and $\hat{I}_{pz}\hat{I}_{qz}$ terms. These

first-order time-independent homonuclear coupling terms can contribute to the residual line width under FSLG decoupling and thus lead to an additional line broadening.

In principle, the second-order effective Hamiltonian during FSLG decoupling contains three types of commutator cross-terms. However, contributions from chemical-shift cross-terms ($\hat{\bar{\mathcal{H}}}_{I \otimes I}$) only contain one-spin operators and will thus lead to an

555 additional effective field and will only weakly influence the residual line width under FSLG by changing the direction or magnitude of the effective field. The same is true for the one-spin component of the dipolar-dipolar cross terms ($\hat{\bar{\mathcal{H}}}_{II \otimes II}$). This leaves only two sources of coupling terms in the second-order effective Hamiltonian: the three-spin contribution of dipolar-dipolar cross-terms and commutators between chemical-shift and dipolar terms. Out of these two, the former will be most relevant for the residual linewidth as dipolar couplings are generally much larger than typical chemical shifts. The corresponding $p_{\mu\chi\xi}^{(n_1,n_2)}$

scaling factors for the three-spin contribution in $\hat{\bar{\mathcal{H}}}_{II \otimes II}$ were computed according to Eq. (35). Because the effective fields generated by the modulations of the rf-field amplitude and phase are small (s. Fig. 12), $\nu = 0$ was excluded from the summation in order to avoid near-resonance conditions and the norm

$$\bar{p}_3^{(n_1,n_2)} = \sqrt{\sum_{\mu,\chi,\xi} \left| p_{\mu\chi\xi}^{(n_1,n_2)} \right|^2} \tag{40}$$

was computed to characterize the strength of the three-spin coupling terms. Logarithmic contour plots for the resulting $\bar{p}_3^{(n_1,n_2)}$

for C1–C4 in the 3.2 mm MAS probe are shown in Fig. 14. Scaling factors are shown for $n_1 = 1$ (Fig. 14a) and $n_1 = 2$ (Fig. 14b) and all possible values of the index $n_2$. It can be seen that time-dependent rf modulations (C2, C3 and C4) increase the





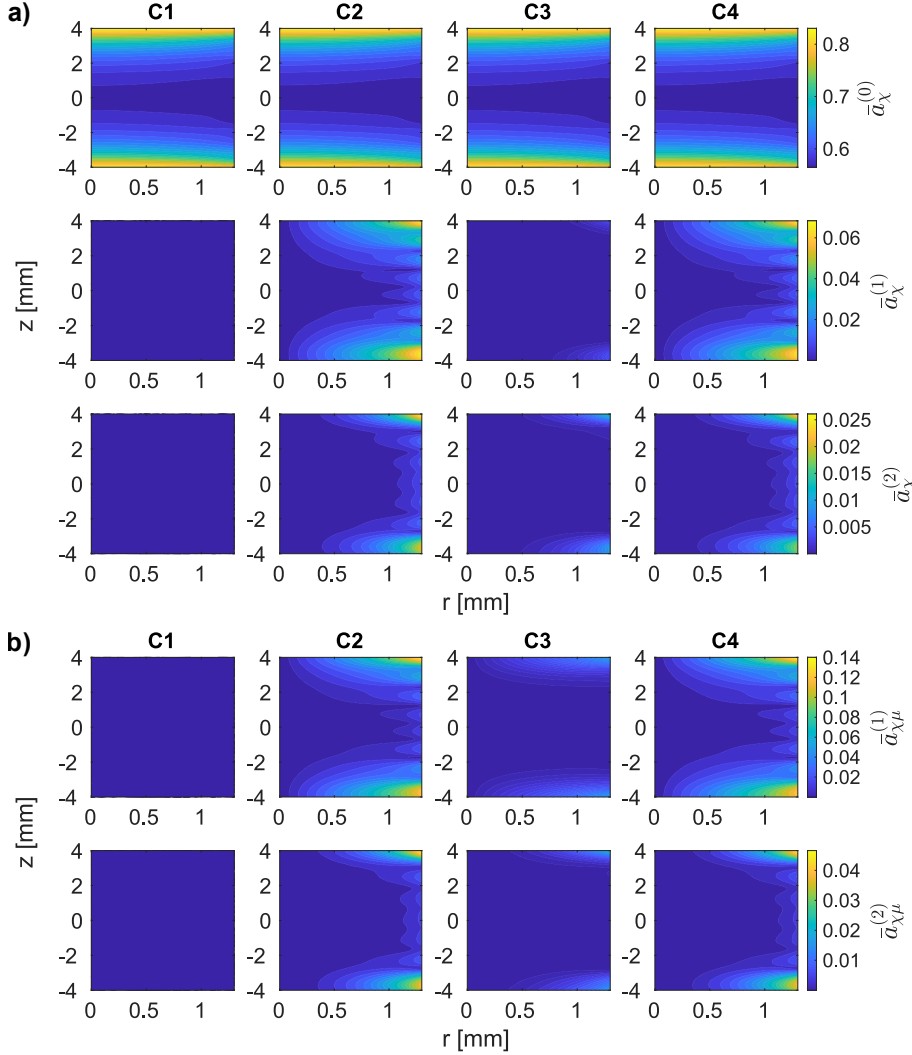

**Figure 13.** Norm of the scaling factors of the chemical shift terms $\bar{a}_\chi^{(k)}$ (for $k = 0, 1, 2$, a) and the homonuclear dipolar coupling terms $\bar{a}_{\chi\mu}^{(k)}$ (for $k = 1, 2$, b) in the first-order effective Hamiltonian for FSLG decoupling for C1–C4. Coefficients were extracted from interaction-frame trajectories of single-spin operators for a nominal rf-field strength of 125 kHz along the magic angle and a MAS frequency of 12.5 kHz in a 3.2 mm MAS probe. a) The scaling of the isotropic chemical shift ($k = 0$) is unaffected by the time-dependent rf-field modulations (no difference between C1–C4) and is close to the ideal value of $\cos\theta_m \approx 0.577$ in the centre of the rotor, where the rf-field amplitude corresponds to the nominal rf-field strength. Scaling factors increase towards the rotor edges where the rf-field amplitude is significantly lower leading to a smaller tilt angle. Additional non-zero coefficients for $k = 1$ and $2$ terms are obtained when amplitude modulations are taken into account (C2 and C4). b) Amplitude modulations (C2 and C4) lead to non-zero scaling factors and thus to the reintroduction of dipolar couplings in areas where strong modulations occur. No significant effects are observed for the two remaining cases.





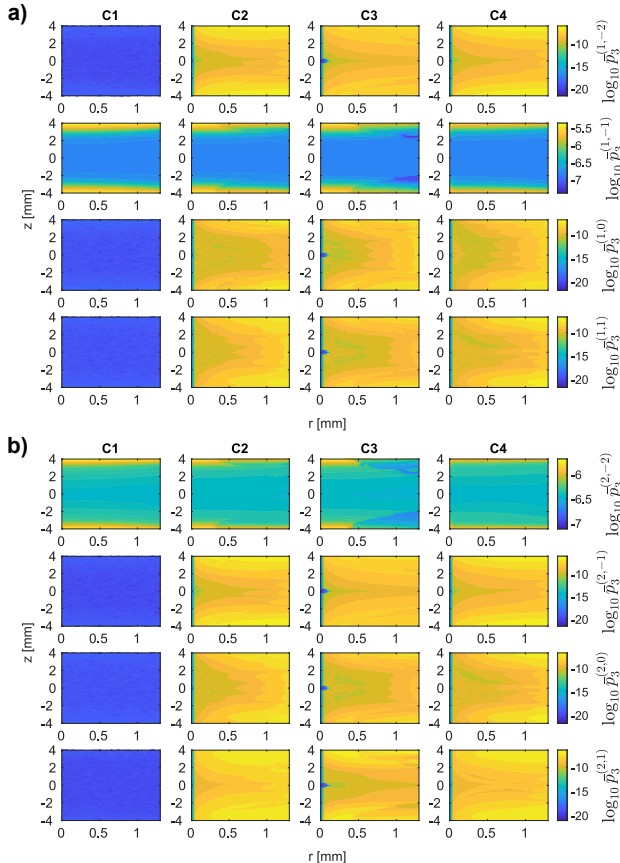

**Figure 14.** Magnitude of the scaling factors of the three-spin contribution to the dipolar-dipolar cross-terms in the second-order effective Hamiltonian during FSLG decoupling with an rf-field strength of 125 kHz along the magic angle at a MAS frequency of 12.5 kHz in a 3.2 mm MAS probe. The resulting magnitudes are shown for $n_1 = 1$ (a) and $n_1 = 2$ (b) for all possible $n_2$ values with a logarithmic scale. The largest scaling factors are obtained for combinations where $n_1 + n_2 = 0$, however no difference between C1–C4 is observed. For other $n_1, n_2$ pairs, significantly larger scaling factors result when rf modulations are present (C2, C3 and C4). Nevertheless, they remain several orders of magnitude smaller than the first-order contributions.

magnitude of the second-order cross terms significantly compared to static rf amplitude and phase offsets alone (C1). However, the observed scaling factors are still negligible compared to the magnitude of the first-order terms and no significant effect on the linewidth would be expected. In all four cases, substantially higher scaling factors are obtained for pairs of indices where

$n_1 + n_2 = 0$. These increase strongly towards the edges of the rotor along the rotor axis but since no difference between the cases C1–C4 is observed they do not seem to be influenced by the radial part of the rf inhomogeneity.

The analysis of the scaling factors of the terms contributing to the effective Hamiltonian up to second order suggests that the static part of the rf inhomogeneity has a significant influence on the isotropic chemical-shift scaling and also leads to stronger



second-order contributions. However, the overall magnitude of these second-order terms remains small compared to first-order contributions. Time-dependent rf amplitude modulations have pronounced first-order effects and lead to the reintroduction of anisotropic chemical-shift and dipolar coupling terms that will potentially cause line broadening. No such effects were observed for phase modulations.

## 6 Conclusion and Outlook

Magic-angle spinning in combination with inhomogeneous radial rf fields leads to a time-dependent modulation of the rf-field amplitude and phases. We have investigated the effect of these time-dependent rf fields on some common solid-state NMR pulse sequences using numerical simulations and an analytical approach based on Floquet theory. In none of the investigated building blocks used in solid-state NRM experiments could we find significant effects from such time-dependent rf fields. In nutation spectra, two distinct families of sidebands arising due to rf-field amplitude and rf-field phase modulations, respectively, were observed in simulated as well as experimental spectra. The intensity of these sidebands can help characterize the strength of the modulations and thus gives insights into the radial contribution to the rf-field inhomogeneity for a given MAS probe. In the polarization-transfer sequences like Hartmann-Hahn cross polarization, REDOR, and C7, only minor effects were observed that will most likely be of no consequence for experimental implementations. In all these sequences, the static rf-field inhomogeneity over the sample volume played a much larger role and leads to significant performance degradation.

In simulations of homonuclear FSLG decoupling, considerable line broadening was observed for rf-field amplitude modulations. Floquet analysis of the effective Hamiltonian up to second-order revealed that this broadening is most likely due to the reintroduction of homonuclear coupling terms to first order caused by the MAS modulation of the rf-field amplitude. However, no experimental characterization of this effect was possible as the experimentally obtained line widths were not limited by the homonuclear decoupling. Overall, the results presented in this work suggest that the influence of the MAS modulation of the rf-field amplitude and phase in many pulse sequences is small and thus negligible for typical experimental implementations. Moreover, they manifest themselves in areas of the sample space close to the rotor edges and can thus be reduced by physical or virtual sample restriction. Nevertheless, these modulations can become relevant in the development of new pulse sequences based on optimal-control strategies and should be taken into account in their development in order to increase their robustness towards rf inhomogeneity and enlarge the NMR-responsive sample volume.

*Data availability.* The data will be made available through the ETH library data services.

*Supplement.* The supplement related to this article is available online at: https://doi.org/10.5194/mr-0-1-2021-supplement.



*Author contributions.* ME designed the research. ZT provided data about rf-field inhomogeneity in probes. KA carried out all measurements and simulations with some help from ME. All the authors discussed and interpreted the results and were involved in writing the manuscript.

*Competing interests.* The authors declare that they have no conflict of interest.

*Acknowledgements.* We would like to thank Perunthiruthy K. Madhu, Kaustubh Mote, and Johannes Hellwagner for insightful discussions about theory and experimental implementation of homonuclear decoupling. Beat H. Meier and Alexander Barnes are acknowledged for providing measurement time for the project.

*Financial support.* This research has been supported by the Schweizerischer Nationalfonds zur Förderung der Wissenschaftlichen Forschung
(grant no. 200020_188988) and the Czech Science Foundation (grant no. 20-00166J).





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
