# Peer review of "Effects of radial radio-frequency field inhomogeneity on MAS solid-state NMR experiments"

_Magnetic Resonance, 2021_

## Author Response (AR1)

| 1      | All points raised by the two reviewers have been answered in the response to the review   |
|--------|-------------------------------------------------------------------------------------------|
|        | reports and are again summarized here. Changes to the manuscript are indicated in te      |
|        | responses and in the summary document.                                                    |
| 2      |                                                                                           |
| 3      |                                                                                           |
| 4      | Reviewer 1                                                                                |
| 5      |                                                                                           |
| 6      | Thanks for the positive comments and the carefull reading of the manuscript.              |
| 7      | On The this sensitive they may well be wight. He ways they may have missed at least one   |
| 8      | Q: In this conclusion they may well be right. However they may have missed at least one   |
|        | prior report in the titerature. In 1988 (doi.org/10.1002/1jch.198800039) rotary resonance |
|        | recoupling experiments were reported in which an unanticipated peak appeared in the       |
|        | discussion around equation 27 postulates a spinning induced periodic phase modulation     |
|        | of the radio frequency field due to inhomogeneity in the field direction of the came      |
|        | type analysed in the current namer (this was actually 3 years before the Goldman article  |
|        | cited by the authors as the seminal reference). Simulations including a highly simulified |
|        | model of this phase modulation did reproduce the central peak (figure 6). It would be     |
|        | interesting to know whether the far more sophisticated calculations performed by the      |
|        | authors verify – or disprove – this finding.                                              |
| 9      |                                                                                           |
| 10     | A: We were not aware of the effect of time-modulated rf fields in R3 recoupling and would |
|        | like to thank you for bringing this to our attention. This is guite interesting and we    |
|        | have incorporated it in the introduction. We have modified the relevant part on page 2    |
|        | to:                                                                                       |
| 11     |                                                                                           |
| 12     | "In solid-state NMR under magic-angle spinning (MAS) conditions, the radial component of  |
|        | the rf field gets modulated by time (Levitt et al., 1988; Tekely and Goldman, 2001;       |
|        | Goldman and Tekely, 2001; Tošner et al., 2017) leading to further potential complications |
|        | in the experiments. Such MAS-induced time-dependent radio-frequency fields could give     |
|        | rise to additional or modified resonance conditions or to other changes in the effective  |
|        | Hamiltonians generated by the pulse sequence. The importance of such time-dependent terms |
|        | was first described in rotary-resonance recoupling (Levitt et al., 1988) where it leads   |
|        | to changes in the observed line shape. Besides the appearance of additional sidebands in  |
|        | cross-polarization experiments (rekely and Goldman, 2001; Goldman and Tekely, 2001),      |
|        | magnetization in MLEV16 sequences under MAS (Piotto et al. 2001) there have been very     |
|        | few studies of the effects of such modulations of the amplitude and phase of the rf field |
|        | caused by MAS rotation."                                                                  |
| 13     |                                                                                           |
| 14     | We have simulated the R3 experiment and see the same effect of an additional peak         |
|        | appearing in the center of the powder pattern identical what is shown in the original     |
|        | reference. Are more detailed discussion of static and time-dependent inhomogeneity has    |
|        | been added to the SI section S4 and is referenced at the end of the spin-lock section of  |
|        | the main paper.                                                                           |
| 15     |                                                                                           |
| 16     | Q: Apart from this comment which might require some minor changes to the paper the only   |
|        | criticism I have for this excellent piece of work is that the use of radians as the phase |
|        | unit makes some of the figures needlessly hard to interpret. I suggest they plot the      |
|        | phase in units of degrees for ease of interpretation – or if they really do not like      |
|        | degrees, plot phase divided by pi, or similar.                                            |
| 10     | A: Since we have no special preference for radians, we have changed the units for the     |
| 10     | hase to degrees in figures 1 a                                                            |
|
10 |                                                                                           |
| 20     |                                                                                           |
| 21     | Reviewer 2:                                                                               |
| 22     |                                                                                           |
| 23     | We would like to thank the reviewer for the very careful reading of the manuscript and    |
|        | the useful suggestions to improve the presentation.                                       |
| 24     |                                                                                           |
| 25     | Q: First, in the section on nutation experiments, the authors attribute the lower         |

**/Users/maer/mirror/frame/papers/radial\_rf\_fields/response\_all.txt Saved: 07.06.21, 08:32:02**

25... intensity of the sidebands caused by amplitude modulations (compared to the phase modulation sidebands) partially to the lower magnitude of the amplitude modulations .... themselves. However, the magnitudes of both modulations do not seem too different when .... looking at Fig. 2 and taking the different scales of amplitude and phase into account. .... The second explanation provided by the authors (the broadening of the sideband by axial .... rf inhomogeneity) seems more likely. This could be tested in a simulation that only .... includes the radial amplitude modulation and not the axial inhomogeneity profile, which would remove the broadening of the sideband. .... 26 A: We have looked into this in a bit more detail and this is correct. We have deleted the 27 sentence on line 277: "The reduced intensity of these amplitude-modulation sidebands can .... be explained by the lower magnitude of amplitude modulations in comparison to phase .... modulations (see Fig. 2)." The reason is really the distribution of the sidebands over a .... larger spectral range which we have also confirmed in simulations as suggested by the .... reviewer. .... 28 Q: Second, in the section on cross-polarization, the difference of the rf field 29 inhomogeneity of the two rf fields across the sample is held responsible for the ••• restriction of the active sample volume. But the rf fields on the two channels do not .... need to have different rf profiles, the mere existence of rf inhomogeneity is sufficient .... to make the matching condition in Eq. 37 impossible to attain within the whole sample at the same time. 30 31 A: We agree with the reviewer that the presence of rf-field inhomogeneity alone will lead to a mismatch in parts of the probe. We have rephrased the relevant text after EQ. (37) ... to: "Due to the rf-field inhomogeneity across the sample, this condition cannot be .... fulfilled simultaneously in the entire sample volume and only certain parts of the sample .... will participate in the polarization transfer thus decreasing the resulting signal intensity." 32 Q: Fig. 2 may benefit from two changes. The first would put the panels showing z = +4 mm 33 and z = -4 mm on the same vertical scales (they are very similar already, but not the same). The second would add another set of panels showing the largest radius of each .... axial position on the same scale to illustrate the difference in magnitude along the ... rotor axis. .... 34 A: We have adjusted the scaling of the two panels in Fig.2 but we have not added a 35 separate panel with the largest radius. The reason for this is that the figure would become too small to recognise any details. In addition, the shape of the rf inhomogeneity .... is not something that we investigated but only copied and used from previous work. .... 36 Q: In Fig. 3, vertical scales would be especially useful on the insets showing the 37 sidebands caused by amplitude modulation, since they are barely visible in the main .... graphs, making an estimation of the scale all but impossible. .... 38 A: We agree that this is a good idea and have added the vertical axis to the insets. 39 40 Q: The meaning of the arrows in Fig. 11 is described in section 4.2, but not in the 41 figure caption, where I was first looking to find out what they meant. 42 A: We have added a sentence to the figure caption that explains the meaning of the 43 arrows: "The arrows indicate the positions of the carrier frequency." .... 44 45 Q: The paper by Tošner et al. from 2017 does not include any actual optimal control 46 calculations, as suggested by the citation in line 43 (their tm-SPICE pulses were .... presented in 2018). .... 47 A: We have corrected the citation for the Tosner 2017 and 2018 papers. 48 49 Q: The insets in panels b) and d) of Fig. 3 look exactly the same despite the relevant 50 parts of the main graphs looking different, has there been an accidental duplication?

```
51
  A: There was indeed a mixup and one of the insets was duplicated and wrong. Thanks a lot
52
   for spotting this. It has been updated in the revised version of the manuscript.
...
53
  Q: In line 437, Fig. 9c/d is referenced, but the authors likely mean to reference Fig.
54
   9e/f.
....
55
  A: corrected
56
57
  Q: The first sentence of the caption to Fig. 9 makes it look like the simulations for all
58
   panels were done at 150 MHz, which, according to the text and a later sentence in the
....
   same caption, is not the case.
....
59
  A: corrected
60
61
  Q: In the last sentence of the caption to Fig. 10, the line splitting is placed in panel
62
   d) instead of c), and it can be attributed.
....
63
  A: corrected
64
65
  Q: "NRM" in line 583 is probably supposed to read "NMR".
66
67
  A: corrected
68
69
70
```